# AdaMSS: Adaptive Multi-Subspace Approach for Parameter-Efficient Fine-Tuning

**Jingjing Zheng**[1,2,3,6]
jjzheng233@gmail.com

**Wanglong Lu**[4]
Wanglong.Lu@nasdaq.com

**Yiming Dong**[1]
yimingdong_ml@outlook.com

**Chaojie Ji**[2,3,6]
chaojiej@math.ubc.ca

**Yankai Cao**[2,3,5,†]
yankai.cao@ubc.ca

**Zhouchen Lin**[1,7,8,†]
zlin@pku.edu.cn

[1]State Key Lab of General AI, School of Intelligence Science and Technology, Peking University
[2]Institute of Applied Mathematics, The University of British Columbia
[3]Centre for AI Decision-Making and Action, The University of British Columbia
[4]AI Analytics Team, Nasdaq, St. John's, NL, Canada
[5]Department of Chemical and Biological Engineering, The University of British Columbia
[6]Department of Mathematics, The University of British Columbia
[7]Institute for Artificial Intelligence, Peking University
[8]Pazhou Laboratory (Huangpu), Guangzhou, Guangdong, China

## Abstract

In this paper, we propose AdaMSS, an adaptive multi-subspace approach for parameter-efficient fine-tuning of large models. Unlike traditional parameter-efficient fine-tuning methods that operate within a large single subspace of the network weights, AdaMSS leverages subspace segmentation to obtain multiple smaller subspaces and adaptively reduces the number of trainable parameters during training, ultimately updating only those associated with a small subset of subspaces most relevant to the target downstream task. By using the lowest-rank representation, AdaMSS achieves more compact expressiveness and finer tuning of the model parameters. Theoretical analyses demonstrate that AdaMSS has better generalization guarantee than LoRA, PiSSA, and other single-subspace low-rank-based methods. Extensive experiments across image classification, natural language understanding, and natural language generation tasks show that AdaMSS achieves comparable performance to full fine-tuning and outperforms other parameter-efficient fine-tuning methods in most cases, all while requiring fewer trainable parameters. Notably, on the `ViT-Large` model, AdaMSS achieves 4.7% higher average accuracy than LoRA across seven tasks, using just 15.4% of the trainable parameters. On `RoBERTa-Large`, AdaMSS outperforms PiSSA by 7% in average accuracy across six tasks while reducing the number of trainable parameters by approximately 94.4%. These results demonstrate the effectiveness of AdaMSS in parameter-efficient fine-tuning. The code for AdaMSS is available at `https://github.com/jzheng20/AdaMSS`.

## 1 Introduction

With the successful application of large pre-trained models in natural language processing and computer vision, the parameter-efficient fine-tuning (PEFT) methods have gradually become key

39th Conference on Neural Information Processing Systems (NeurIPS 2025).

---

[†]Corresponding authors

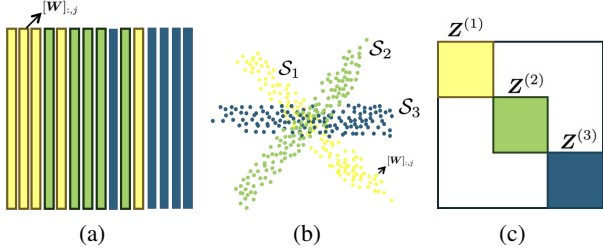

Figure 1: (a) Weight matrix $\boldsymbol{W}$; (b) Multi-subspaces structure of the columns of $\boldsymbol{W}$ (*i.e.*, $[\boldsymbol{W}]_{:,j}$); (c) The lowest-rank representation of $\boldsymbol{W}$, where each block $\boldsymbol{Z}^{(k)}$ serves as a new representation for the columns lying in the subspace $\mathcal{S}_k$.

strategies for efficiently adapting these models. Among these PEFT methods, LoRA (Low-Rank Adaptation) [1] has gained particular attention, as it adopts learnable low-rank structures to represent weight changes, significantly reducing memory requirements. Following LoRA, numerous low-rank adaptation methods have since been developed [2, 3, 4, 5, 6, 7, 8, 9, 10, 11].

Despite sharing the common goal of PEFT, these methods primarily differ in how they impose low-rank assumptions. For instance, LoRA [1] and AdaLoRA [3] assume that weight changes $\Delta \boldsymbol{W} \in \mathbb{R}^{n \times n}$ are low-rank and represent them using the product of two smaller matrices, $\boldsymbol{A} \in \mathbb{R}^{n \times r}$ and $\boldsymbol{B} \in \mathbb{R}^{r \times n}$, where $r \ll n$. Alternatively, PiSSA [2] assumes that the original weight matrix $\boldsymbol{W}$ is approximately low-rank and achieves efficient fine-tuning by training its principal components. A further line of research is represented by methods such as LoRA-GA [5], which hypothesize that the gradient of the loss function with respect to the weights—*i.e.*, $\nabla \mathcal{L}(\boldsymbol{W})$—exhibits low-rank structure.

These approaches are typical single-subspace methods, as they rely on the assumption that weight changes, weights, or gradients lie in, or approximately lie in, a single low-dimensional subspace. However, this assumption inevitably leads to a fundamental trade-off between limited expressiveness and parameter efficiency: approximating the weights with a single low-dimensional subspace with a small $r$ limits the model's capacity for adaptation, while increasing the subspace dimension $r$ improves expressiveness but results in a substantial rise in the number of trainable parameters and memory usage.

To address the dilemma between limited expressiveness and parameter efficiency, we propose an **Ada**ptive **M**ulti-**S**ubspace approach (AdaMSS) that enables finer tuning of model parameters and has stronger expressiveness. As we will illustrate in this work, the columns of the weight matrix are approximately distributed across multiple linear subspaces (Figure 1 (a)-(b) provides a conceptual illustration of multi-subspaces structure of the network weights).

The key features of this work are summarized as follows:

- **Compact Expressiveness**: By leveraging the observation that the network weights are approximately located in multiple subspaces, AdaMSS uses a lowest-rank representation for the network weights, which exhibit an approximate block-diagonal structure (see Figure 1 (c)). This design leads to a more compact representation while preserving both global and local low-rank structure from the original weights (see Property 1), which is essential for ensuring effective adaptation across multiple subspaces.

- **Generalization Guarantee**: We theoretically show that, under the same rank assumption $r$, the proposed AdaMSS achieves a lower Gaussian complexity bound and is therefore expected to exhibit a stronger generalization capability compared to single-subspace-based methods. Theorem 1 formalizes an upper bound on the expected loss of AdaMSS, providing its generalization guarantee.

- **Multi-Subspace-Based Adaptive Budget Allocation**: Thanks to the multi-subspace structure of the network weights, we can adaptively freeze the parameters associated with subspaces of lower importance, and eventually updates only those associated with a small subset of subspaces relevant to the target downstream task, where the importance of each subspace is evaluated during training.

Experimental results demonstrate the effectiveness of AdaMSS for PEFT across diverse tasks and show that it outperforms existing methods in most cases while significantly reducing the number of trainable parameters. For example, compared with LoRA, AdaMSS achieves 4.7% higher average accuracy on `ViT-Large` [12] and 8.5% higher accuracy on GSM8K [13] with `LLaMA 2-7B` [14], using only 13.63% and 1.25% of the trainable parameters, respectively. On `RoBERTa-Large` [15], AdaMSS achieves comparable performance to LoRA while using just 5.62% of the trainable parameters. These results suggest that multi-subspace-based fine-tuning offers a promising direction for improving the trade-off between expressiveness and parameter efficiency.

## 2 Preliminaries

In this section, we introduce the notations used throughout the paper, and provide preliminaries and a brief review of existing literature relevant to our study. Unless otherwise specified, we use $a$, $\boldsymbol{a}$, and $\boldsymbol{A}$ to denote scalars, vectors, and matrices, respectively. The transpose of $\boldsymbol{A}$ is denoted by $\boldsymbol{A}^\top$. We use $[\boldsymbol{A}]_{i,j}$, $[\boldsymbol{A}]_{i,:}$, and $[\boldsymbol{A}]_{:,j}$ to represent the $(i,j)$-th element, the $i$-th row vector, and the $j$-th column vector of $\boldsymbol{A}$, respectively.

### 2.1 Low-Rank Representation for Subspace Segmentation

To get the lowest-rank presentation for $n$ given samples [2] $[\boldsymbol{M}]_{:,1}, [\boldsymbol{M}]_{:,2}, \cdots, [\boldsymbol{M}]_{:,n} \in \mathbb{R}^d$, Liu *et al.* proposed the Low-Rank Representation (LRR) model [16] expressed as follows:

$$[\boldsymbol{Z}^\star, \boldsymbol{E}^\star] = \min_{\boldsymbol{Z},\boldsymbol{E}} \mathrm{rank}(\boldsymbol{Z}) + \lambda\|\boldsymbol{E}\|_\Diamond \qquad s.t.\ \boldsymbol{M} = \boldsymbol{M}\boldsymbol{Z} + \boldsymbol{E}, \tag{1}$$

where $\boldsymbol{Z}$ is known as the coefficient matrix, $\mathrm{rank}(\boldsymbol{Z})$ denotes the rank of matrix $\boldsymbol{Z} \in \mathbb{R}^{n \times n}$, *i.e.*, the number of singular values of $\boldsymbol{Z}$, $\boldsymbol{M} \in \mathbb{R}^{d \times n}$ represents the matrix composed of the $n$ samples, $\boldsymbol{E}$ represents the residual component (*i.e.*, noise and redundancy in the samples). The norm $\|\cdot\|_\Diamond$ is used to quantify the magnitude of the nonzero entries of $\boldsymbol{E}$, and its specific definition depends on the prior knowledge about $\boldsymbol{E}$. The problem (1) can be approximated by a convex formulation and solved iteratively [16], and it is computationally expensive. Fortunately, when $\boldsymbol{E} = \boldsymbol{0}$, problem (1) has a closed-form solution, given by $\boldsymbol{Z}^\star = \boldsymbol{V}\boldsymbol{V}^\top$, where $\boldsymbol{V} \in \mathbb{R}^{n \times \mathrm{rank}(\boldsymbol{M})}$ is obtained through the skinny SVD of $\boldsymbol{M}$, *i.e.*, $\boldsymbol{M} = \boldsymbol{U}\boldsymbol{S}\boldsymbol{V}^\top$.

Assuming the samples $[\boldsymbol{M}]_{:,1}, [\boldsymbol{M}]_{:,2}, \cdots, [\boldsymbol{M}]_{:,n}$ drawn from a union of linear subspaces $\{\mathcal{S}_k\}_{k=1}^K$, where the subspace number $K$ and $\{\mathcal{S}_k\}_{k=1}^K$ are unknown, the goal of subspace segmentation is to simultaneously assign the samples into their respective subspaces. When the subspaces $\{\mathcal{S}_k\}_{k=1}^K$ are independent of each other, samples from different subspaces cannot be expressed as linear combinations of each other. Consequently, $\boldsymbol{Z}^\star$ becomes a block-diagonal matrix. To solve the subspace segmentation problem, Liu *et al.* determine the subspace number $K$ and to obtain the segmentation results based on $\boldsymbol{Z}^\star$ [16]. This LLR-based subspace segmentation method is widely applied in clustering tasks[16, 17, 18, 19].

## 3 Multi-Subspace-Based Adaptation

Let $\boldsymbol{W}_0 \in \mathbb{R}^{d \times n}$ denote the pretrained weight matrix of a network layer, where $d$ and $n$ represent the input and output dimensions of the layer, respectively. The full fine-tuning (FF) of the network is formalized as

$$\Delta\boldsymbol{W}^\star = \arg\min_{\Delta\boldsymbol{W} \in \mathbb{R}^{d \times n}} \mathcal{L}(\boldsymbol{W}_0 + \Delta\boldsymbol{W}), \tag{2}$$

where $\mathcal{L}$ denotes the loss function, $\Delta\boldsymbol{W} \in \mathbb{R}^{d \times n}$ is the update on $\boldsymbol{W}_0$, and $\boldsymbol{W} = \boldsymbol{W}_0 + \Delta\boldsymbol{W}$ is the corresponding updated weights. In this section, we aim to find a compact representation of $\boldsymbol{W}_0$ via subspace segmentation [16], which has both low-rank and strictly block-diagonal structure. By doing so, instead of directly updating $\boldsymbol{W}_0$ as FF, we perform incremental updates within this compact representation.

---

[2] In this work, the *samples* here does not refer to layer input $\boldsymbol{x}$.

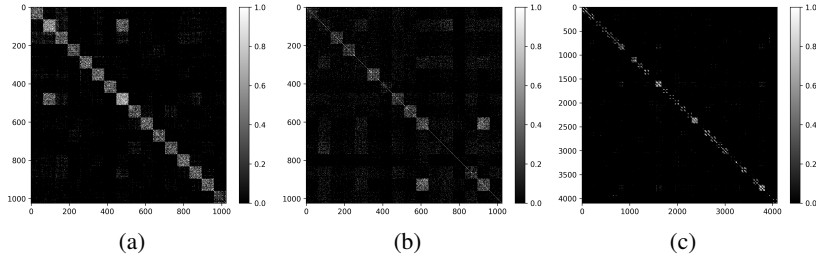

|  (a)  |  (b)  |  (c)  |

Figure 2: Illustration of multiple subspaces structure in pretrained network weights: an approximate block-diagonal structure of $\boldsymbol{Z}_0^\star$ from the first layer of the pretrained (a) `ViT-Large` (query), (b) `Roberta-Large` (query), and (c) `LLaMA 2-7B` (query).

### 3.1 Motivation: Multi-Subspace Structure of the Network Weights

Motivated by PiSSA [2], we focus on updating the principle components of $\boldsymbol{W}_0$ (denoted as $\hat{\boldsymbol{W}}_0$), and decompose the pretrained weight $\boldsymbol{W}_0$ as $\boldsymbol{W}_0 = \hat{\boldsymbol{W}}_0 + \boldsymbol{W}_{res}$, where $\boldsymbol{W}_{res} := \boldsymbol{W}_0 - [\boldsymbol{U}_0]_{:,1:R}[\boldsymbol{S}_0]_{1:R,1:R}[\boldsymbol{V}_0]_{:,1:R}^\top$ for a given $R$, and $\hat{\boldsymbol{W}}_0 = [\boldsymbol{U}_0]_{:,1:R}[\boldsymbol{S}_0]_{1:R,1:R}[\boldsymbol{V}_0]_{:,1:R}^\top$ is the truncated SVD of $\boldsymbol{W}_0$. We apply the LRR to obtain a lowest-rank representation of $\hat{\boldsymbol{W}}_0$. Considering computational complexity, we adapt the LRR with $\boldsymbol{E} = \boldsymbol{0}$:

$$\boldsymbol{Z}_0^\star = \underset{\boldsymbol{Z}_0 \in \mathbb{R}^{n \times n}}{\arg\min} \operatorname{rank}(\boldsymbol{Z}_0) \qquad s.t.\ \hat{\boldsymbol{W}}_0 = \hat{\boldsymbol{W}}_0 \boldsymbol{Z}_0. \tag{3}$$

This formulation uses the coefficient matrix $\boldsymbol{Z}_0$ to linearly represent the columns of $\hat{\boldsymbol{W}}_0 \in \mathbb{R}^{d \times n}$. By analyzing $\boldsymbol{Z}_0^\star$, we observe that it exhibits an approximate block-diagonal structure (see Figure 2 for $R = 100$), indicating that the columns of $\hat{\boldsymbol{W}}_0 = \hat{\boldsymbol{W}}_0 \boldsymbol{Z}_0^\star$ are grouped into different clusters, where columns within the same cluster are approximately linear combinations of each other. In other words, the row space of $\boldsymbol{W}_0$ can be approximated by multiple smaller low-rank subspaces. By leveraging the approximate block-diagonal structure and low-rank properties of $\boldsymbol{Z}_0^\star$, we drive a more compact representation of $\boldsymbol{W}_0$ than $\hat{\boldsymbol{W}}_0$.

### 3.2 A Compact Representation By Subspace Segmentation

In the previous subsection, we analyzed the multi-subspace structure of $\boldsymbol{W}_0$ and obtained a compact representation $\boldsymbol{Z}_0^\star$, which exhibits an approximate block-diagonal structure. We now derive a well-organized and strictly block-diagonal representation of $\boldsymbol{W}_0$ by subspace segmentation.

Leveraging subspace segmentation and the multi-subspace structure, the row space of $\boldsymbol{W}_0$ can be approximately segmented into $K$ smaller low-dimensional subspaces. We assume that the columns in $\boldsymbol{W}_0 = [\boldsymbol{W}_0^{(1)}, \boldsymbol{W}_0^{(2)}, \cdots, \boldsymbol{W}_0^{(K)}]$ are well-organized and partitioned based on the segmentation results, where each block $\boldsymbol{W}_0^{(k)} \in \mathbb{R}^{d \times n_k}$ contains the columns assigned to the $k$-th subspace. Accordingly, $\hat{\boldsymbol{W}}_0$, $\boldsymbol{Z}_0^\star$, and $\boldsymbol{W}_{res}$ are also divided into $K$ blocks, leading to the following decomposition

$$\begin{aligned}
&[\boldsymbol{W}_0^{(1)}, \boldsymbol{W}_0^{(2)}, \cdots, \boldsymbol{W}_0^{(K)}] \\
&= [\hat{\boldsymbol{W}}_0^{(1)}, \hat{\boldsymbol{W}}_0^{(2)}, \cdots, \hat{\boldsymbol{W}}_0^{(K)}]\Big(\operatorname{diag}\Big((\boldsymbol{Z}_0^\star)^{(1)}, (\boldsymbol{Z}_0^\star)^{(2)}, \cdots, (\boldsymbol{Z}_0^\star)^{(K)}\Big) + [\boldsymbol{E}_0^{(1)}, \boldsymbol{E}_0^{(2)}, \cdots, \boldsymbol{E}_0^{(K)}]\Big) \\
&\quad + [\boldsymbol{W}_{res}^{(1)}, \boldsymbol{W}_{res}^{(2)}, \cdots, \boldsymbol{W}_{res}^{(K)}],
\end{aligned} \tag{4}$$

where $\boldsymbol{Z}_0^\star = [[\boldsymbol{Z}_0^\star]^{(1)}, [\boldsymbol{Z}_0^\star]^{(2)}, \cdots, [\boldsymbol{Z}_0^\star]^{(K)}]$ is decomposed into a sum of a strictly block-diagonal matrix $\operatorname{diag}\Big((\boldsymbol{Z}_0^\star)^{(1)}, (\boldsymbol{Z}_0^\star)^{(2)}, \cdots, (\boldsymbol{Z}_0^\star)^{(K)}\Big)$ and a noise matrix $[\boldsymbol{E}_0^{(1)}, \boldsymbol{E}_0^{(2)}, \cdots, \boldsymbol{E}_0^{(K)}]$, and each block $(\boldsymbol{Z}_0^\star)^{(k)} \in \mathbb{R}^{n_k \times n_k}$ is the $k$-th diagonal block of $\boldsymbol{Z}_0^\star$.

Define $\bar{\boldsymbol{W}}_{res}^{(k)} := \hat{\boldsymbol{W}}_0^{(k)} \boldsymbol{E}_0^{(k)} + \boldsymbol{W}_{res}^{(k)}$, we have

$$\boldsymbol{W}_0^{(k)} = \hat{\boldsymbol{W}}_0^{(k)} (\boldsymbol{Z}_0^\star)^{(k)} + \bar{\boldsymbol{W}}_{res}^{(k)} \tag{5}$$

**Algorithm 1** Initialization of AdaMSS

---

**Input:** $\boldsymbol{W}_0$, $R$, and $r_k$.
**Output:** Estimated $K$, $\{\boldsymbol{A}^{(k)}\}_{k=1}^{K}$, and the initialized values of $\{\boldsymbol{B}^{(k)}\}_{k=1}^{K}$ and $\{\boldsymbol{C}^{(k)}\}_{k=1}^{K}$.

**Step 1**: Compute the skinny SVD of $\boldsymbol{W}_0$: $\boldsymbol{W}_0 = \boldsymbol{U}_0 \boldsymbol{S}_0 \boldsymbol{V}_0^\top$;
**Step 2**: Obtain $\hat{\boldsymbol{W}}_0$ by $\hat{\boldsymbol{W}}_0 = [\boldsymbol{U}_0]_{:,1:R}[\boldsymbol{S}_0]_{1:R,1:R}[\boldsymbol{V}_0]_{:,1:R}^\top$ for given $R$;
**Step 3**: Obtain $\boldsymbol{Z}_0^\star$ by $\boldsymbol{Z}_0^\star = [\boldsymbol{V}_0]_{:,1:R}[\boldsymbol{V}_0]_{:,1:R}^\top$;     ▷ Steps 1-3 are derived from Section 3.1.
**Step 4**: Determine (or estimate) the number of subspaces $K$, and apply clustering to assign the the columns of $\boldsymbol{W}_0$ to $K$ subspaces [16], as detailed in Algorithm 2 (see Appendix C);
**Step 5**: Obtain the matrix blocks $\{\boldsymbol{W}_0^{(k)}\}_{k=1}^{K}$, $\{\hat{\boldsymbol{W}}_0^{(k)}\}_{k=1}^{K}$, and $\{(\boldsymbol{Z}_0^\star)^{(k)}\}_{k=1}^{K}$ by the segmentation results;     ▷ Steps 4-5 are derived from Section 3.2.
**Step 6**: For each $k$, compute the skinny SVD of $\hat{\boldsymbol{W}}_0^{(k)}$: $\hat{\boldsymbol{W}}_0^{(k)} = \boldsymbol{U}_{\hat{\boldsymbol{W}}_0}^{(k)} \boldsymbol{S}_{\hat{\boldsymbol{W}}_0}^{(k)} (\boldsymbol{V}_{\hat{\boldsymbol{W}}_0}^{(k)})^\top$;
**Step 7**: Perform the QR decomposition of $\boldsymbol{S}_{\hat{\boldsymbol{W}}_0}^{(k)} (\boldsymbol{V}_{\hat{\boldsymbol{W}}_0}^{(k)})^\top (\boldsymbol{Z}_0^\star)^{(k)}$: $\boldsymbol{S}_{\hat{\boldsymbol{W}}_0}^{(k)} (\boldsymbol{V}_{\hat{\boldsymbol{W}}_0}^{(k)})^\top (\boldsymbol{Z}_0^\star)^{(k)} = \boldsymbol{Q}^{(k)} \boldsymbol{R}^{(k)}$;
**Step 8**: Set $\boldsymbol{A}^{(k)} = \boldsymbol{U}_{\hat{\boldsymbol{W}}_0}^{(k)}$, initialize $\boldsymbol{B}^{(k)} = \boldsymbol{Q}^{(k)}$ and $\boldsymbol{C}^{(k)} = \boldsymbol{0}$.     ▷ Steps 6-8 are derived from Section 3.3.

---

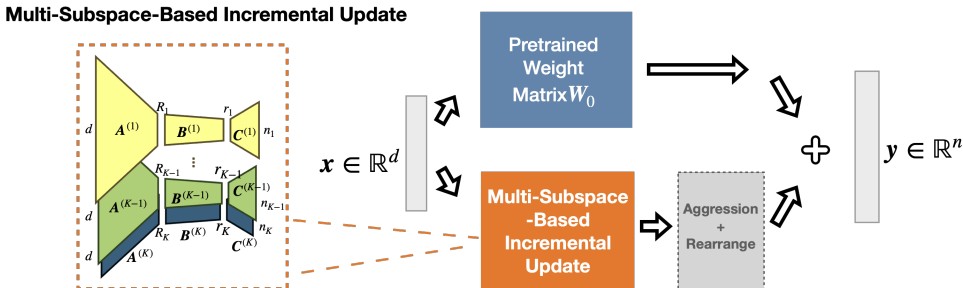

Figure 3: Illustration of the proposed Multi-Subspace-Based Adaptation framework.

for each $k$, where $(\boldsymbol{Z}_0^\star)^{(k)} \in \mathbb{R}^{n_k \times n_k}$ is used to approximately represents $\boldsymbol{W}_0^{(k)}$. We thus obtain

$$[\boldsymbol{W}_0^{(1)}, \boldsymbol{W}_0^{(2)}, \cdots, \boldsymbol{W}_0^{(K)}] = [\hat{\boldsymbol{W}}_0^{(1)}, \hat{\boldsymbol{W}}_0^{(2)}, \cdots, \hat{\boldsymbol{W}}_0^{(K)}]\text{diag}\left((\boldsymbol{Z}_0^\star)^{(1)}, (\boldsymbol{Z}_0^\star)^{(2)}, \cdots, (\boldsymbol{Z}_0^\star)^{(K)}\right)$$
$$+ [\bar{\boldsymbol{W}}_{res}^{(1)}, \bar{\boldsymbol{W}}_{res}^{(2)}, \cdots, \bar{\boldsymbol{W}}_{res}^{(K)}], \tag{6}$$

and a new representation of $\boldsymbol{W}_0$:

$$\text{diag}\left((\boldsymbol{Z}_0^\star)^{(1)}, (\boldsymbol{Z}_0^\star)^{(2)}, \cdots, (\boldsymbol{Z}_0^\star)^{(K)}\right). \tag{7}$$

The following properties hold for this representation.

**Property 1.** *If the subspaces are independent (*i.e.*, $\text{rank}(\boldsymbol{Z}_0^\star) = \sum_{k=1}^{K} \text{rank}([\boldsymbol{Z}_0^\star]^{(k)}))$, then*
*(1) (Global structure-preserving)* $\sum_{k=1}^{K} \text{rank}(\hat{\boldsymbol{W}}_0^{(k)}) = \sum_{k=1}^{K} \text{rank}((\boldsymbol{Z}_0^\star)^{(k)}) = \text{rank}(\hat{\boldsymbol{W}}_0)$;
*(2) (Local structure-preserving)* $\text{rank}((\boldsymbol{Z}_0^\star)^{(k)}) = \text{rank}(\hat{\boldsymbol{W}}_0^{(k)})$ *for* $k = 1, 2, \cdots, K$.

*Proof.* The corresponding proof is given in the Appendix A.     □

Property 1 ensures both global and local low-rank structure preservation by the representation of $\boldsymbol{W}_0$. Specifically, the total rank across all subspaces remains consistent with that of the original weight matrix (Property 1 (1)), while each subspace individually preserves its intrinsic low-rank structure (Property 1 (2)). These properties are critical for maintaining original model abilities and enabling reliable adaptation across multiple subspaces.

## 3.3 Multi-Subspace-Based Incremental Update

In the previous subsection, we presented a representation of $\boldsymbol{W}_0$ based on subspace segmentation, as shown in (7). In this subsection, we propose a multi-subspace-based adaptation strategy. Rather than updating $\boldsymbol{W}_0$ directly, the proposed adaptation performs incremental updates for each $(\boldsymbol{Z}_0^\star)^{(k)}$.

We introduce an incremental update $\Delta\boldsymbol{Z}^{(k)}$ (with rank $r_k$) for each $(\boldsymbol{Z}_0^\star)^{(k)}$ in (6), and compute the updated weight block $\boldsymbol{W}^{(k)}$ as

$$\boldsymbol{W}^{(k)} = \hat{\boldsymbol{W}}_0^{(k)}((\boldsymbol{Z}_0^\star)^{(k)} + \Delta\boldsymbol{Z}^{(k)}) + \bar{\boldsymbol{W}}_{res}^{(k)} = \boldsymbol{W}_0^{(k)} + \hat{\boldsymbol{W}}_0^{(k)}\Delta\boldsymbol{Z}^{(k)}, \tag{8}$$

where the second equality follows from (5). Since the computation involves multiplication with the large matrix $\hat{\boldsymbol{W}}_0^{(k)}$, calculating the gradient with respect to $\Delta\boldsymbol{Z}^{(k)}$ can be computationally expensive. To address this, we decompose $\hat{\boldsymbol{W}}_0^{(k)}$ by the skinny SVD: $\hat{\boldsymbol{W}}_0^{(k)} = \boldsymbol{U}_{\hat{\boldsymbol{W}}_0}^{(k)}\boldsymbol{S}_{\hat{\boldsymbol{W}}_0}^{(k)}(\boldsymbol{V}_{\hat{\boldsymbol{W}}_0}^{(k)})^\top$. Let $R_k := \mathrm{rank}(\hat{\boldsymbol{W}}_0^{(k)})$ and $\boldsymbol{A}^{(k)} := \boldsymbol{U}_{\hat{\boldsymbol{W}}_0}^{(k)} \in \mathbb{R}^{d\times R_k}$. Then, the update becomes:

$$\boldsymbol{W}^{(k)} = \boldsymbol{W}_0^{(k)} + \boldsymbol{A}^{(k)}\boldsymbol{S}_{\hat{\boldsymbol{W}}_0}^{(k)}(\boldsymbol{V}_{\hat{\boldsymbol{W}}_0}^{(k)})^\top\Delta\boldsymbol{Z}^{(k)}. \tag{9}$$

Since $\mathrm{rank}(\boldsymbol{S}_{\hat{\boldsymbol{W}}_0}^{(k)}(\boldsymbol{V}_{\hat{\boldsymbol{W}}_0}^{(k)})^\top\Delta\boldsymbol{Z}^{(k)}) \leq r_k$, we introduce two trainable matrices $\boldsymbol{B}^{(k)} \in \mathbb{R}^{R_k\times r_k}$ and $\boldsymbol{C}^{(k)} \in \mathbb{R}^{r_k\times n_k}$, and reparameterize the $k$-th updated weight block as:

$$\boldsymbol{W}^{(k)} = \boldsymbol{W}_0^{(k)} + \boldsymbol{A}^{(k)}\boldsymbol{B}^{(k)}\boldsymbol{C}^{(k)}. \tag{10}$$

This formulation yields a low-rank update (*i.e.*, $\boldsymbol{A}^{(k)}\boldsymbol{B}^{(k)}\boldsymbol{C}^{(k)}$) within each subspace. The overall framework is illustrated in Figure 3.

To ensure that the initial value of $\boldsymbol{W}^{(k)}$ remains consistent with $\boldsymbol{W}_0^{(k)}$, we initialize $\boldsymbol{C}^{(k)}$ in (10) as $\boldsymbol{0}$. The matrix $\boldsymbol{B}^{(k)} \in \mathbb{R}^{R_k\times r_k}$ is initialized either as an orthogonal matrix, as described in Steps 7-8 of the Algorithm 1. The complete initialization procedure of AdaMSS is presented in Algorithm 1. The parameter count and computational complexity analysis of the proposed method, along with comparisons to LoRA and PiSSA, are provided in the Appendices D and E, respectively.

## 4 Multi-Subspace-Based Adaptive Budget Allocation

In this section, we introduce an adaptive multi-subspace-based budget allocation mechanism. By integrating this strategy with the Multi-Subspace-Based Adaptation framework proposed in the previous section, we derive the final method, AdaMSS.

Thanks to the multi-subspace structure of the weights, we can perform adaptive budget allocation during training by calculating the importance score [20] for each $\boldsymbol{H}^{(k)} = [\boldsymbol{B}^{(k)};(\boldsymbol{C}^{(k)})^\top] \in \mathbb{R}^{(R_k+n_k)\times r_k}$ directly. Following [20], the importance score $s^{(t)}(\cdot)$ is defined as

$$s^{(t)}(\boldsymbol{H}^{(k)}) = \frac{1}{r_k(R+n_k)}\sum_{i,j}\overline{I}^{(t)}([\boldsymbol{H}^{(k)}]_{i,j})\cdot\overline{U}^{(t)}([\boldsymbol{H}^{(k)}]_{i,j}), \tag{11}$$

where

$$\overline{I}^{(t)}([\boldsymbol{H}^{(k)}]_{i,j}) = \beta_1\overline{I}^{(t-1)}([\boldsymbol{H}^{(k)}]_{i,j}) + (1-\beta_1)I^{(t)}([\boldsymbol{H}^{(k)}]_{i,j}),$$

$$\overline{U}^{(t)}([\boldsymbol{H}^{(k)}]_{i,j}) = \beta_2\overline{U}^{(t-1)}([\boldsymbol{H}^{(k)}]_{i,j}) + (1-\beta_2)|I^{(t)}([\boldsymbol{H}^{(k)}]_{i,j}) - \overline{I}^{(t)}([\boldsymbol{H}^{(k)}]_{i,j})|, \tag{12}$$

$$I([\boldsymbol{H}^{(k)}]_{ij}) = |[\boldsymbol{H}^{(k)}]_{ij}\nabla_{[\boldsymbol{H}^{(k)}]_{ij}}\mathcal{L}|,$$

and $t$ denotes the training step. This importance score measures how consistently and significantly each $\boldsymbol{H}^{(k)}$ contributes to loss reduction during training.

At each update step, we rank all $\boldsymbol{H}^{(k)}$ by their importance scores $s^{(t)}(\boldsymbol{H}^{(k)})$, retain the top-$K_t$ for training, and freeze the rest. The value of $K_t$ is gradually reduced from $K$ to $K_{\mathrm{target}}$ following a smooth cubic decay schedule (*i.e.*, with exponent $\rho = 3$) over training steps. The adaptive budget allocation mechanism is stopped when the number of trainable $\boldsymbol{H}^{(k)}$ for $k = 1, 2, \cdots, K$ falls below

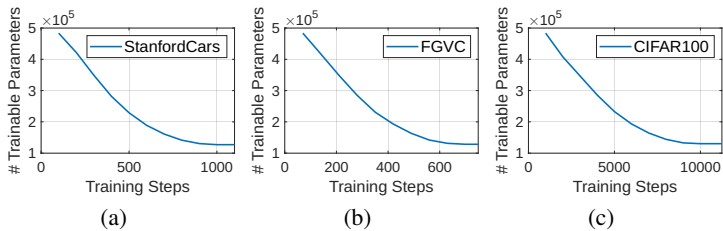

(a)          (b)          (c)

Figure 4: The adaptive change in the number of trainable parameters of AdaMSS during the training process.

$K_{\text{target}}$. Figure 4 presents examples on the StanfordCars[3], FGVC[3], and CIFAR100[3] tasks, illustrating the proposed adaptive budget allocation strategy, which gradually reduces the number of trainable parameters during the training process.

## 5 Analysis of AdaMSS

In the previous section, we introduced AdaMSS that is based on the following reparametrization:

$$\boldsymbol{W} = \boldsymbol{W}_0 + \boldsymbol{ABC}, \tag{13}$$

where $\boldsymbol{A} = [\boldsymbol{A}^{(1)}, \boldsymbol{A}^{(2)}, \cdots, \boldsymbol{A}^{(K)}] \in \mathbb{R}^{d \times R}$, $\boldsymbol{B} = \text{diag}(\boldsymbol{B}^{(1)}, \boldsymbol{B}^{(2)}, \cdots, \boldsymbol{B}^{(K)}) \in \mathbb{R}^{R \times \sum_{k=1}^{K} r_k}$, $\boldsymbol{C} = \text{diag}(\boldsymbol{C}^{(1)}, \boldsymbol{C}^{(2)}, \ldots, \boldsymbol{C}^{(K)}) \in \mathbb{R}^{\sum_{k=1}^{K} r_k \times n}$, $(\boldsymbol{A}^{(k)})^\top \boldsymbol{A}^{(k)} = \boldsymbol{I}$ and $\text{rank}(\boldsymbol{C}^{(k)}) \leq r_k$ for $k = 1, 2, \cdots, K$.

We establish the upper bound of expected loss by AdaMSS in Lemma 1 and Theorem 1.

As our discussions given in the Appendix B.2, the upbound of the Gaussian complexity for LoRA and its single subspace variants (such as AdaLoRA, PiSSA and LoRA-GA) is given by $\mathcal{L}(\phi) \hat{R} B \sqrt{\frac{\text{rank}(\Delta \boldsymbol{W}) n}{m}}$ for $\|\Delta \boldsymbol{W}\|_2 \leq B$. If $\sum_{k=1}^{K} r_k = \text{rank}(\Delta \boldsymbol{W})$, applying Cauchy–Schwarz inequality and using $\sum_{k=1}^{K} n_k = n$, we have $\sum_{k=1}^{K} \sqrt{r_k n_k} \leq \sqrt{\text{rank}(\Delta \boldsymbol{W}) n}$. Therefore, AdaMSS attains a lower Gaussian complexity bound when $B_k \leq B$, and can be expected to have a stronger generalization capability than single subspace-based fine-tuning methods, as indicated by Theorem 1. These results can be further extended to deep Lipschitz neural networks by applying Maurer's Gaussian complexity chain rule [21].

**Lemma 1.** *[Gaussian complexity of the AdaMSS for a shallow Lipschitz neural network] For the class of spectrally bounded shallow Lipschitz network*

$$\mathcal{F}_{\text{AdaMSS}} = \{ f_{\boldsymbol{W}}(\boldsymbol{x}) = \phi(\boldsymbol{x}\boldsymbol{W}) \mid \boldsymbol{W} = \boldsymbol{W}_0 + \boldsymbol{ABC},$$
$$(\boldsymbol{A}^{(k)})^\top \boldsymbol{A}^{(k)} = \boldsymbol{I}, \ \|\boldsymbol{B}^{(k)} \boldsymbol{C}^{(k)}\|_2 \leq B_k \}, \tag{14}$$

*the Gaussian complexity of this function class is upper bounded as follows:*

$$\hat{G}_S(\mathcal{F}_{\text{AdaMSS}}) \leq \mathcal{L}(\phi) \sum_{k=1}^{K} \hat{R} B_k \sqrt{\frac{r_k n_k}{m}},$$

*where $n = \sum_{k=1}^{K} n_k$, $n_k$ denotes the width of the weight matrix $\boldsymbol{C}^{(k)}$, $\mathcal{L}(\phi)$ is Lipschitz constant for function $\phi$, $\boldsymbol{X} = [\boldsymbol{x}_1^\top, \ldots, \boldsymbol{x}_m^\top] \in \mathbb{R}^{d \times m}$ for the samples $\{\boldsymbol{x}_i\}_{i=1}^{m}$, and $\max_{i=1,2,\cdots,m} \|\boldsymbol{x}_i\| \leq \hat{R}$.*

*Proof.* The corresponding proof is given in the Appendix B.1. $\qquad \square$

**Theorem 1.** *Let $g(\cdot)$ be a $\mathcal{L}(g)$-Lipschitz loss function from $(f_{\mathbf{w}}(\mathbf{x}), \mathbf{y})$ to $[0, 1]$, where $f_{\mathbf{w}} \in \mathcal{F}_{\text{AdaMSS}}$ and $(\boldsymbol{x}, \boldsymbol{y}) \in \mathbb{X} \times \mathbb{Y}$, $\mathbb{X} \subseteq \mathbb{R}^d$ and $\mathbb{Y}$ are feature space and output space, respectively. For any $\delta > 0$, the following holds with probability at least $1 - \delta$ for a randomly chosen i.i.d. samples $\mathbb{S} = \{(\mathbf{x}_i, \mathbf{y}_i)\}_{i=1}^{m}$:*

$$\mathbb{E}[g(f_{\mathbf{w}}(\mathbf{x}), \mathbf{y})] \leq \frac{1}{m} \sum_{i=1}^{m} g(f_{\mathbf{w}}(\mathbf{x}_i), \mathbf{y}_i) + \sqrt{\mathcal{L}(g)\pi} \mathcal{L}(\phi) \sum_{k=1}^{K} \hat{R} B_k \sqrt{\frac{r_k n_k}{m}} + \sqrt{\frac{9 \log \frac{2}{\delta}}{2m}}.$$

---
[3]https://huggingface.co/datasets/Multimodal-Fatima

*Proof.* Combining Lemma 1 with Vector-valued Gaussian complexity Generalization Bound Theorem [22], we obtain the generalization bound for AdaMSS. □

# 6 Experiments

In this section, we present comparative experiments to evaluate the effectiveness of AdaMSS across tasks in image classification (IC), natural language understanding (NLU), and natural language generation (NLG). To better understand the contribution of each component, we define a variant of AdaMSS without the proposed adaptive budget allocation mechanism, referred to as AdaMSS$_{\text{base}}$, for clarity of comparison. We report the average number of trainable parameters per epoch in our experimental results, as the number of trainable parameters in AdaMSS is adaptively reduced during the training. Since our focus is on low-rank-based adaptation, we compare AdaMSS not only with baseline methods such as full fine-tuning (FF) and linear probing (LP), which only tune the classification head, but also with several low-rank-based PEFT methods, including LoRA [1], DyLoRA [23], AdaLoRA [3], PiSSA [2], LoRETTA [10], and LoRA-PRO [6]. For each method, we prioritize using hyperparameters (*e.g.*, rank $r$) recommended by the authors. For a fair comparison, we replicate the experimental setups in [1, 2, 24]. A complete list of all hyperparameters and settings is provided in the Appendix H. Ablation studies are given in Appendix G. All experiments were performed with Python version 3.12.3. The best results are highlighted in **bold**, and the top three results are underlined.

## 6.1 Image Classification

Table 1: The performance of different fine-tuning methods on various image classification datasets for the `ViT-Base` model and `ViT-Large` model, averaged over 5 random seeds, where the hyperparameter $r_k$ in AdaMSS$_{\text{base}}$ and AdaMSS corresponds to the assumed rank of $\Delta Z^{(k)}$ and is set to the same value for all $k$.

| Model | Method | # Trainable Parameters | Accuracy (%) | | | | | | | |
|---|---|---|---|---|---|---|---|---|---|---|
| | | | OxfordPets | StanfordCars | CIFAR10 | EuroSAT | FGVC | RESISC45 | CIFAR100 | Avg. |
| ViT-Base | LP | - | $90.28_{\pm 0.4}$ | $25.76_{\pm 0.3}$ | $96.41_{\pm 0.0}$ | $88.72_{\pm 0.1}$ | $17.44_{\pm 0.4}$ | $74.22_{\pm 0.1}$ | $84.28_{\pm 0.1}$ | 68.15 |
| | FF | 85.8M | $93.14_{\pm 0.4}$ | $\underline{79.78}_{\pm 1.2}$ | $\mathbf{98.92}_{\pm 0.1}$ | $\mathbf{99.05}_{\pm 0.1}$ | $\underline{54.84}_{\pm 1.2}$ | $\mathbf{96.13}_{\pm 0.1}$ | $92.38_{\pm 0.1}$ | $\underline{87.74}$ |
| | LoRA ($r=16$) | 581K | $93.19_{\pm 0.4}$ | $45.38_{\pm 0.4}$ | $\underline{98.78}_{\pm 0.1}$ | $98.44_{\pm 0.2}$ | $25.16_{\pm 0.2}$ | $92.70_{\pm 0.2}$ | $92.02_{\pm 0.1}$ | 77.95 |
| | PiSSA ($r=8$) | 313K | $93.84_{\pm 0.3}$ | $78.43_{\pm 0.5}$ | $98.74_{\pm 0.0}$ | $\underline{98.67}_{\pm 0.1}$ | $51.56_{\pm 1.8}$ | $93.81_{\pm 1.8}$ | $\mathbf{93.31}_{\pm 0.2}$ | 86.90 |
| | PiSSA ($r=1$) | $\underline{55K}$ | $93.83_{\pm 0.1}$ | $60.29_{\pm 0.3}$ | $98.7_{\pm 0.0}$ | $98.47_{\pm 0.1}$ | $29.61_{\pm 0.2}$ | $92.87_{\pm 0.2}$ | $91.98_{\pm 0.2}$ | 80.82 |
| | LoRA-PRO | 313K | $\underline{94.03}_{\pm 0.1}$ | $72.12_{\pm 0.4}$ | $98.77_{\pm 0.1}$ | $98.65_{\pm 0.1}$ | $43.39_{\pm 0.7}$ | $93.66_{\pm 0.2}$ | $\underline{92.54}_{\pm 0.1}$ | 84.74 |
| | WeGeFT | $\underline{49K}$ | $92.71_{\pm 0.2}$ | $76.18_{\pm 0.2}$ | $98.46_{\pm 0.1}$ | $94.92_{\pm 6.9}$ | $51.82_{\pm 0.9}$ | $93.03_{\pm 0.2}$ | $91.46_{\pm 0.2}$ | 85.51 |
| | LoRETTA ($r=5$) | 57K | $93.39_{\pm 0.4}$ | $74.15_{\pm 0.8}$ | $98.73_{\pm 0.1}$ | $\underline{98.67}_{\pm 0.1}$ | $48.86_{\pm 0.5}$ | $93.36_{\pm 0.1}$ | $91.87_{\pm 0.1}$ | 85.57 |
| | AdaMSS$_{\text{base}}$ ($r_k=3$) | 125K | $94.02_{\pm 0.3}$ | $\mathbf{81.55}_{\pm 0.4}$ | $\underline{98.81}_{\pm 0.05}$ | $98.7_{\pm 0.06}$ | $\mathbf{56.95}_{\pm 0.6}$ | $94.18_{\pm 0.1}$ | $92.13_{\pm 0.1}$ | $\mathbf{88.05}$ |
| | AdaMSS ($r_k=3$) | 59K | $\mathbf{94.23}_{\pm 0.1}$ | $\underline{80.44}_{\pm 0.2}$ | $98.69_{\pm 0.04}$ | $98.59_{\pm 0.09}$ | $54.45_{\pm 0.3}$ | $\underline{94.03}_{\pm 0.2}$ | $91.91_{\pm 0.1}$ | $\underline{87.47}$ |
| | AdaMSS$_{\text{base}}$ ($r_k=1$) | $\mathbf{42K}$ | $93.91_{\pm 0.2}$ | $78.98_{\pm 0.2}$ | $98.71_{\pm 0.07}$ | $98.64_{\pm 0.07}$ | $53.2_{\pm 0.4}$ | $93.62_{\pm 0.08}$ | $91.90_{\pm 0.1}$ | 86.99 |
| ViT-Large | LP | - | $91.11_{\pm 0.3}$ | $37.91_{\pm 0.3}$ | $97.78_{\pm 0.0}$ | $92.64_{\pm 0.1}$ | $24.62_{\pm 0.2}$ | $82.02_{\pm 0.1}$ | $84.28_{\pm 0.1}$ | 72.91 |
| | FF | 303.3M | $94.43_{\pm 0.6}$ | $\mathbf{88.90}_{\pm 0.3}$ | $\underline{99.15}_{\pm 0.1}$ | $\mathbf{99.04}_{\pm 0.1}$ | $\mathbf{68.25}_{\pm 1.6}$ | $\mathbf{96.43}_{\pm 0.1}$ | $\underline{93.58}_{\pm 0.2}$ | $\mathbf{91.40}$ |
| | LoRA ($r=16$) | 1.57M | $\underline{94.82}_{\pm 0.1}$ | $73.25_{\pm 0.4}$ | $\underline{99.13}_{\pm 0.0}$ | $98.63_{\pm 0.1}$ | $42.32_{\pm 1.6}$ | $94.71_{\pm 0.3}$ | $\mathbf{94.87}_{\pm 0.1}$ | 85.40 |
| | PiSSA ($r=8$) | 835K | $94.04_{\pm 0.4}$ | $84.19_{\pm 0.7}$ | $\underline{99.13}_{\pm 0.0}$ | $98.79_{\pm 0.0}$ | $59.81_{\pm 0.6}$ | $94.99_{\pm 0.2}$ | $92.42_{\pm 0.1}$ | 89.05 |
| | PiSSA ($r=1$) | $\underline{147K}$ | $93.98_{\pm 0.3}$ | $83.04_{\pm 0.3}$ | $99.04_{\pm 0.0}$ | $98.71_{\pm 0.0}$ | $56.72_{\pm 0.6}$ | $94.64_{\pm 0.2}$ | $93.25_{\pm 0.1}$ | 88.48 |
| | LoRA-PRO | 835K | $94.67_{\pm 0.1}$ | $83.57_{\pm 0.3}$ | $\mathbf{99.20}_{\pm 0.1}$ | $98.81_{\pm 0.1}$ | $57.71_{\pm 0.5}$ | $95.12_{\pm 0.1}$ | $\underline{93.53}_{\pm 0.1}$ | 88.94 |
| | WeGeFT | 204K | $94.49_{\pm 0.2}$ | $83.96_{\pm 0.1}$ | $99.06_{\pm 0.1}$ | $98.34_{\pm 0.1}$ | $60.82_{\pm 0.5}$ | $94.49_{\pm 0.3}$ | $92.69_{\pm 0.2}$ | 89.12 |
| | LoRETTA ($r=5$) | $\mathbf{132K}$ | $78.28_{\pm 0.3}$ | $68.44_{\pm 0.3}$ | $98.80_{\pm 0.2}$ | $98.68_{\pm 0.1}$ | $65.30_{\pm 0.6}$ | $94.53_{\pm 0.1}$ | $93.28_{\pm 0.1}$ | 82.86 |
| | AdaMSS$_{\text{base}}$ ($r_k=3$) | 483K | $94.74_{\pm 0.1}$ | $85.40_{\pm 0.3}$ | $99.11_{\pm 0.0}$ | $\underline{98.93}_{\pm 0.06}$ | $65.30_{\pm 0.6}$ | $95.32_{\pm 0.1}$ | $93.51_{\pm 0.1}$ | $\underline{90.33}$ |
| | AdaMSS ($r_k=3$) | 241K | $\mathbf{94.87}_{\pm 0.1}$ | $85.24_{\pm 0.3}$ | $99.12_{\pm 0.0}$ | $98.93_{\pm 0.1}$ | $64.31_{\pm 0.4}$ | $95.2_{\pm 0.2}$ | $93.22_{\pm 0.1}$ | $\underline{90.13}$ |
| | AdaMSS$_{\text{base}}$ ($r_k=1$) | $\underline{178K}$ | $94.58_{\pm 0.1}$ | $83.71_{\pm 0.2}$ | $99.08_{\pm 0.05}$ | $\underline{98.85}_{\pm 0.1}$ | $59.27_{\pm 0.8}$ | $94.68_{\pm 0.3}$ | $93.43_{\pm 0.2}$ | 89.09 |

We evaluate all methods on IC using the widely adopted Vision Transformer (ViT) [12], a prevalent foundation model in computer vision, across seven public datasets: OxfordPets[4], StanfordCars[3], CIFAR10[3], EuroSAT[5], FGVC[3], RESISC45[6], and CIFAR100[3]. The number of training epochs is set as 10. As shown in Table 1, AdaMSS achieves higher average accuracy than other PEFT methods, including LP, LoRA, PiSSA, LoRA-PRO, and LoRETTA, and performs comparable accuracy with FF on the seven image classification datasets while using significantly fewer trainable parameters. More specifically, with the ViT-Base model, AdaMSS (59K) achieves average accuracy comparable to PiSSA ($r=8$), using only 19.5% of its tranable parameters, and achieves 6.5% higher accuracy than PiSSA ($r=1$) at a similar parameter budget. This is because that AdaMSS's multi-subspace design allows it to capture richer features with fewer parameters. On the ViT-large model, AdaMSS achieves 4.7% higher average accuracy than LoRA, while requiring only with 15.4% of the trainable

---

[4]https://huggingface.co/datasets/timm/oxford-iiit-pet

[5]https://huggingface.co/datasets/timm/eurosat-rgb

[6]https://huggingface.co/datasets/timm/resisc45

parameters, demonstrating AdaMSS's strong parameter efficiency and transfer performance in image classification tasks.

## 6.2 Natural Language Understanding

Table 2: The performance of different fine-tuning methods on six datasets of the GLUE benchmark for the `RoBERTa-Base` model and `RoBERTa-Large` model, averaged over 5 random seeds.

| Model | Method | # Trainable Parameters | SST-2 Acc. | MRPC Acc. | CoLA MCC | QNLI Acc. | RTE Acc. | STS-B PCC | Avg. |
|---|---|---|---|---|---|---|---|---|---|
| | FF | 125M | 94.8 | **90.2** | 63.6 | 92.8 | 78.7 | 91.2 | 85.2 |
| | LoRA | 0.3M | **95.1**$_{\pm 0.2}$ | 89.7$_{\pm 0.7}$ | 63.4$_{\pm 1.2}$ | **93.3**$_{\pm 0.3}$ | 78.4$_{\pm 0.8}$ | **91.5**$_{\pm 0.2}$ | 85.2 |
| | AdaLoRA | 0.3M | 94.5$_{\pm 0.2}$ | 88.7$_{\pm 0.5}$ | 62.0$_{\pm 0.6}$ | 93.1$_{\pm 0.2}$ | **81.0**$_{\pm 0.6}$ | 90.5$_{\pm 0.2}$ | 85.0 |
| | DyLoRA | 0.3M | 94.3$_{\pm 0.5}$ | 89.5$_{\pm 0.5}$ | 61.1$_{\pm 0.3}$ | 92.2$_{\pm 0.5}$ | 78.7$_{\pm 0.7}$ | 91.1$_{\pm 0.6}$ | 84.5 |
| `RoBERTa-Base` | PiSSA ($r=8$) | 0.3M | 93.9$_{\pm 0.1}$ | 89.3$_{\pm 0.8}$ | 62.1$_{\pm 2.9}$ | 91.3$_{\pm 0.1}$ | 77.3$_{\pm 1.4}$ | 90.5$_{\pm 0.2}$ | 84.1 |
| | LoRA-PRO | 0.3M | 94.2$_{\pm 0.3}$ | 90.1$_{\pm 0.5}$ | 64.3$_{\pm 0.72}$ | 92.0$_{\pm 0.2}$ | 80.2$_{\pm 1.8}$ | 90.9$_{\pm 0.22}$ | **85.3** |
| | LoRA ($r=1$) | 0.055M | 93.7$_{\pm 0.5}$ | 89.2$_{\pm 0.4}$ | 62.3$_{\pm 3.6}$ | 90.6$_{\pm 0.4}$ | 79.5$_{\pm 0.4}$ | 80.8$_{\pm 20.6}$ | 82.7 |
| | PiSSA ($r=1$) | 0.055M | 93.3$_{\pm 0.2}$ | 89.3$_{\pm 0.6}$ | 62.6$_{\pm 1.4}$ | 90.6$_{\pm 0.4}$ | 74.9$_{\pm 1.2}$ | 90.0$_{\pm 0.3}$ | 83.4 |
| | LoRETTA | 0.057M | 94.6$_{\pm 0.7}$ | 88.3$_{\pm 0.7}$ | 61.8$_{\pm 1.3}$ | 92.7$_{\pm 0.2}$ | 75.1$_{\pm 5.3}$ | 90.5$_{\pm 0.1}$ | 83.8 |
| | WeGeFT | 0.049M | 94.1$_{\pm 0.5}$ | 89.5$_{\pm 0.5}$ | 63.5$_{\pm 1.3}$ | 91.2$_{\pm 0.4}$ | 78.6$_{\pm 1.6}$ | 90.5$_{\pm 0.1}$ | 84.6 |
| | AdaMSS$_{base}$ ($r_k=1$) | 0.042M | 94.6$_{\pm 0.2}$ | 89.2$_{\pm 1.0}$ | 64.3$_{\pm 0.9}$ | 92.4$_{\pm 0.1}$ | 77.2$_{\pm 0.7}$ | 90.6$_{\pm 0.1}$ | 84.7 |
| | AdaMSS ($r_k=1$) | **0.032M** | 94.6$_{\pm 0.2}$ | 88.8$_{\pm 1.4}$ | **64.5**$_{\pm 1.1}$ | 92.4$_{\pm 0.1}$ | 77.3$_{\pm 0.7}$ | 90.4$_{\pm 0.1}$ | 84.7 |
| | FF | 356M | 96.4 | 90.9 | 68 | 94.7 | 86.6 | **92.4** | 88.2 |
| | LoRA | 0.8M | 96.2$_{\pm 0.5}$ | 90.2$_{\pm 1.0}$ | 68.2$_{\pm 1.9}$ | **94.8**$_{\pm 0.3}$ | 85.2$_{\pm 1.1}$ | 92.3$_{\pm 0.5}$ | 87.8 |
| | PiSSA ($r=8$) | 0.8M | 95.5$_{\pm 0.2}$ | 86.9$_{\pm 2.6}$ | 61.1$_{\pm 3.4}$ | 92.1$_{\pm 1.7}$ | 56.8$_{\pm 8.2}$ | 91.8$_{\pm 0.4}$ | 80.7 |
| `RoBERTa-Large` | LoRA-PRO | 0.8M | 95.9$_{\pm 0.2}$ | **90.9**$_{\pm 0.4}$ | 66.7$_{\pm 2.0}$ | 93.0$_{\pm 0.5}$ | 60.5$_{\pm 13.5}$ | 92.0$_{\pm 0.1}$ | 83.2 |
| | LoRA ($r=1$) | 0.147M | 95.7$_{\pm 0.4}$ | 88.3$_{\pm 0.7}$ | 62.2$_{\pm 2.4}$ | 93.9$_{\pm 0.2}$ | 82.2$_{\pm 2.5}$ | 78.2$_{\pm 29.7}$ | 83.4 |
| | PiSSA ($r=1$) | 0.147M | 95.2$_{\pm 0.2}$ | 84.9$_{\pm 3.4}$ | 56.6$_{\pm 6.2}$ | 93.4$_{\pm 0.3}$ | 65.9$_{\pm 11.3}$ | 91.3$_{\pm 0.2}$ | 81.2 |
| | LoRETTA | 0.132M | 96.2$_{\pm 0.2}$ | 90.5$_{\pm 0.4}$ | **69.5**$_{\pm 0.6}$ | 94.1$_{\pm 0.9}$ | 53.0$_{\pm 0.5}$ | 92.0$_{\pm 0.2}$ | 82.6 |
| | WeGeFT | 0.065M | 95.0$_{\pm 0.3}$ | 75.7$_{\pm 7.7}$ | 64.0$_{\pm 2.0}$ | 93.7$_{\pm 0.3}$ | 53.6$_{\pm 1.2}$ | 91.4$_{\pm 0.3}$ | 78.9 |
| | AdaMSS$_{base}$ ($r_k=1$) | 0.097M | 96.3$_{\pm 0.2}$ | 90.5$_{\pm 0.3}$ | 68.0$_{\pm 0.9}$ | 94.6$_{\pm 0.1}$ | **87.3**$_{\pm 1.0}$ | 92.0$_{\pm 0.0}$ | 88.1 |
| | AdaMSS ($r_k=1$) | **0.045M** | 96.1$_{\pm 0.0}$ | 90.3$_{\pm 0.5}$ | 67.2$_{\pm 1.2}$ | 94.5$_{\pm 0.1}$ | 87.1$_{\pm 2.1}$ | 91.9$_{\pm 0.0}$ | 87.9 |

We evaluate all methods on the General Language Understanding Evaluation (GLUE) benchmark [25] using the robustly optimized BERT models, *i.e.*, `RoBERTa-Base` and `RoBERTa-Large` [15], for the evaluation. We evaluate the performance of the fine-tuned models using three key metrics: Matthew's correlation coefficient (MCC) for CoLA, Pearson correlation coefficient (PCC) for STS-B, and accuracy (Acc.) for all other tasks. For all methods, the maximum number of training epochs is set to 100 and select the best epoch for each run. Table 2 presents the results of all methods. As shown in the Table 2, on `RoBERTa-Base`, AdaMSS achieves average accuracy comparable to other low-rank-based PEFT methods while using the fewest trainable parameters. For `RoBERTa-Large`, AdaMss outperforms PiSSA, LoRA-PRO, and LoRETTA by around 5% in average accuracy while using fewer parameters, and achieves performance comparable to both LoRA and FF.

## 6.3 Natural Language Generation

In this subsection, we compare AdaMSS with other methods using a range of models, including `LLaMA 2-7B` [14], `Mistral-7B` [26], and `Gemma-7B` [27], on natural language generation (NLG) tasks. We adopt the fixed value $K = 10$ for AdaMSS. All comparison results on accuracy are summarized in Table 3. As the results demonstrate, AdaMSS$^\star$ and AdaMSS$^\star_{base}$ consistently outperforms existing PEFT methods in most cases, while using only a small number of trainable parameters. Notably, AdaMSS achieves state-of-the-art performance on both GSM8K [13], MATH [28] for `LLaMA 2-7B` and `Gemma-7B`, with a clear accuracy margin over all other baselines and using less than 1% of the parameters required for Full FF. For `Mistral-7B`, AdaMSS remains highly competitive and yields the best result on MATH using only 4M trainable parameters.

Table 4 further compares the initialization cost of various PEFT methods on `LLaMA 2-7B`. As shown in the Table 4, AdaMSS incurs a moderate setup cost, significantly lower than LoRETTA but higher than the PiSSA and LoRA-PRO. This reflects the structural complexity of both AdaMSS and LoRETTA, which aim to construct compact representations of network weights: AdaMSS through multi-subspace segmentation, and LoRETTA via tensor decomposition.

## 7 Conclusions and Future Works

In this work, inspired by our observation of the multi-subspace structure in network weights, we proposed AdaMSS, a novel PEFT approach, to address the limitations of single-subspace methods that

Table 3: Comparing different methods on NLG tasks. PEFT methods marked with $\star$ denote tuning of all major projection modules, including $q_{proj}$, $k_{proj}$, $v_{proj}$, $o_{proj}$, $up_{proj}$, $down_{proj}$, and $gate_{proj}$, whereas the default setting only updates $q_{proj}$, and $v_{proj}$.

| Model | Method | Trainable Parameters | GSM8K | MATH |
|---|---|---|---|---|
| LLaMA 2-7B | Full FT | 6738M | 49.05 | 7.22 |
| | LoRA$^\star$ | 320M | 42.30 | 5.50 |
| | PiSSA$^\star$ ($r=8$) | 19M | 44.11 | 5.84 |
| | LoRA-PRO$^\star$ ($r=8$) | 19M | 46.61 | 6.4 |
| | PiSSA ($r=8$) | 4M | 36.39 | 5.35 |
| | LoRETTA ($r=5$) | **0.3M** | 37.86 | 4.6 |
| | AdaMSS$^\star_{base}$ ($r_k=3$) | 4M | **51.10** | **7.57** |
| | AdaMSS$^\star$ ($r_k=3$) | 4M | 50.80 | 7.22 |
| | AdaMSS$_{base}$ ($r_k=3$) | 0.8M | 44.41 | 6.05 |
| Mistral-7B | Full FT | 7242M | 67.02 | 18.6 |
| | LoRA$^\star$ | 168M | 67.70 | 19.68 |
| | PiSSA$^\star$ ($r=8$) | 20M | **71.00** | 20.40 |
| | LoRA-PRO$^\star$ ($r=8$) | 20M | 69.59 | 19.17 |
| | PiSSA ($r=8$) | 3M | 64.26 | 16.87 |
| | LoRETTA ($r=5$) | **0.3M** | 62.6 | 15.6 |
| | AdaMSS$^\star_{base}$ ($r_k=3$) | 4M | 70.71 | **20.44** |
| | AdaMSS$^\star$ ($r_k=3$) | 2M | 70.74 | 19.47 |
| | AdaMSS$_{base}$ ($r_k=3$) | 0.5M | 65.43 | 17.74 |
| Gemma-7B | Full FT | 8538M | 71.34 | 22.74 |
| | LoRA$^\star$ | 200M | 74.90 | 31.28 |
| | PiSSA$^\star$ ($r=8$) | 25M | 75.48 | 29.59 |
| | LoRA-PRO$^\star$ | 25M | 75.90 | 29.25 |
| | PiSSA ($r=8$) | 3M | 71.52 | 27.53 |
| | LoRETTA ($r=5$) | **0.2M** | 70.23 | 26.28 |
| | AdaMSS$^\star_{base}$ ($r_k=3$) | 6M | 75.33 | **29.73** |
| | AdaMSS$^\star$ ($r_k=3$) | 4M | **76.41** | 28.64 |
| | AdaMSS$_{base}$ ($r_k=3$) | 0.7M | 70.86 | 27.38 |

Table 4: Comparing Initialization Costs of Different PEFT Methods on LLama 2-7B.

| Method | PiSSA | LoRA-Pro | LoRETTA | AdaMSS |
|---|---|---|---|---|
| Avg. Time (s) | 2.278 | 2.006 | 12.878 | 8.777 |

often struggle with the dilemma between limited expressiveness and parameter efficiency. Compared to low-rank-based fine-tuning methods, AdaMSS enables more expressive and adaptive parameter updates by leveraging subspace segmentation, thereby achieving both parameter efficiency and strong generalization capability. Comprehensive theoretical analysis and extensive empirical evaluations demonstrate the advantages of AdaMSS. In future work, we aim to extend the multi-subspace perspective beyond fine-tuning to broader areas such as network compression and pruning. We believe that subspace segmentation offers a promising direction for learning compact, disentangled, and efficient representations for network weights, which could benefit model compression and pruning in deep neural networks.

## Acknowledgements

Zhouchen Lin was supported by National Key R&D Program of China (2022ZD0160300), the NSF China (No. 62276004) and the State Key Laboratory of General Artificial Intelligence. Yankai Cao acknowledges funding from discovery program of the Natural Science and Engineering Research Council of Canada (RGPIN-2019-05499) and New Frontiers in Research Fund (NFRFE-2022-00663). The authors also gratefully acknowledge the computing resources and services provided by Digital Research Alliance of Canada (www.alliancecan.ca), and Advanced Research Computing at the University of British Columbia.

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

## A  The Proof of Property 1

*Proof.* From $\boldsymbol{Z}_0^\star = [\boldsymbol{V}_0]_{:,1:R}[\boldsymbol{V}_0]_{:,1:R}^\top$, we know $\mathrm{rank}(\boldsymbol{Z}_0^\star) = \mathrm{rank}(\hat{\boldsymbol{W}}_0)$.

Since $\hat{\boldsymbol{W}}_0^{(k)} = \hat{\boldsymbol{W}}_0^{(k)}(\boldsymbol{Z}_0^\star)^{(k)}$, it follows that $\mathrm{rank}(\hat{\boldsymbol{W}}_0^{(k)}) \leq \mathrm{rank}((\boldsymbol{Z}_0^\star)^{(k)})$. Thus, we have

$$\sum_{k=1}^K \mathrm{rank}(\hat{\boldsymbol{W}}_0^{(k)}) \leq \sum_{k=1}^K \mathrm{rank}((\boldsymbol{Z}_0^\star)^{(k)}) \leq \sum_{k=1}^K \mathrm{rank}([\boldsymbol{Z}_0^\star]^{(k)}) = \mathrm{rank}(\boldsymbol{Z}_0^\star) = \mathrm{rank}(\hat{\boldsymbol{W}}_0).$$

On the other hand, we know that $\mathrm{rank}(\hat{\boldsymbol{W}}_0) \leq \sum_{k=1}^K \mathrm{rank}(\hat{\boldsymbol{W}}_0^{(k)})$. Therefore, we can conclude

$$\sum_{k=1}^K \mathrm{rank}(\hat{\boldsymbol{W}}_0^{(k)}) = \sum_{k=1}^K \mathrm{rank}((\boldsymbol{Z}_0^\star)^{(k)}) = \mathrm{rank}(\hat{\boldsymbol{W}}_0).$$

Since $\mathrm{rank}(\hat{\boldsymbol{W}}_0^{(k)}) \leq \mathrm{rank}((\boldsymbol{Z}_0^\star)^{(k)})$ for all $k$ and $\sum_{k=1}^K \mathrm{rank}(\hat{\boldsymbol{W}}_0^{(k)}) = \sum_{k=1}^K \mathrm{rank}((\boldsymbol{Z}_0^\star)^{(k)})$, it must hold that

$$\mathrm{rank}(\hat{\boldsymbol{W}}_0^{(k)}) = \mathrm{rank}((\boldsymbol{Z}_0^\star)^{(k)}).$$

$\square$

## B  The Proofs of the Results in Section 5

### B.1  The Proof of Lemma 1

*Proof.* By Talagrand's contraction lemma [29], we have

$$\hat{G}_S(\mathcal{F}_{\mathrm{AdaMSS}}) \leq \mathcal{L}(\phi) \frac{1}{m} \mathbb{E}_\gamma \sup_{\boldsymbol{W}} \langle \boldsymbol{W},\, \boldsymbol{X\Gamma} \rangle, \tag{15}$$

where each entry in $\boldsymbol{\Gamma} \in \mathbb{R}^{m \times n}$, *i.e.*, $[\boldsymbol{\Gamma}]_{ij}$, follows standard Gaussian distribution.

Since

$$\mathbb{E}_\gamma[\sup_{\boldsymbol{W}} \langle \boldsymbol{W},\, \boldsymbol{X\Gamma} \rangle] = \mathbb{E}_\gamma[\langle \boldsymbol{W}_0, \boldsymbol{X\Gamma} \rangle] + \mathbb{E}_\gamma[\sup_{\boldsymbol{B},\boldsymbol{C}} \langle \boldsymbol{ABC},\, \boldsymbol{X\Gamma} \rangle]$$
$$= \mathbb{E}_\gamma[\sup_{\boldsymbol{B},\boldsymbol{C}} \langle \boldsymbol{ABC},\, \boldsymbol{X\Gamma} \rangle], \tag{16}$$

we only need to study the upper bound of $\mathbb{E}_\gamma[\sup_{\boldsymbol{B},\boldsymbol{C}} \langle \boldsymbol{ABC},\, \boldsymbol{X\Gamma} \rangle]$.

We divide $\boldsymbol{\Gamma}$ into $K$ blocks according $\{n_k\}_{k=1}^K$, *i.e.*, $\boldsymbol{\Gamma} = [\boldsymbol{\Gamma}^{(1)}, \boldsymbol{\Gamma}^{(2)}, \cdots, \boldsymbol{\Gamma}^{(K)}]$ for $\boldsymbol{\Gamma}^{(k)} \in \mathbb{R}^{m \times n_k}(k = 1, 2, \cdots, K)$. From the Cauchy-Schwarz inequality, we have

$$\sup_{\boldsymbol{B},\boldsymbol{C}} \langle \boldsymbol{ABC}, \boldsymbol{X\Gamma} \rangle = \sum_{k=1}^K \sup_{\boldsymbol{B}^{(k)},\boldsymbol{C}^{(k)}} \langle \boldsymbol{A}^{(k)}\boldsymbol{B}^{(k)}\boldsymbol{C}^{(k)}, \boldsymbol{X\Gamma}^{(k)} \rangle \leq \sum_{k=1}^K \|\boldsymbol{B}^{(k)}\boldsymbol{C}^{(k)}\|_F \|\boldsymbol{X\Gamma}^{(k)}\|_F$$

$$\leq \sum_{k=1}^K \sqrt{r_k} B_k \|\boldsymbol{X\Gamma}^{(k)}\|_F. \tag{17}$$

Using the fact that $\mathbb{E}[|Y|] \leq \sqrt{\mathbb{E}[Y^2]}$ for any random variable $Y$, we obtain

$$\mathbb{E}_\gamma\left[\|\boldsymbol{X\Gamma}^{(k)}\|_F\right] \leq \sqrt{\mathbb{E}_\gamma\left[\|\boldsymbol{X\Gamma}^{(k)}\|_F^2\right]} = \sqrt{\mathrm{Tr}\left(\boldsymbol{X}\boldsymbol{X}^\top \mathbb{E}_\gamma[(\boldsymbol{\Gamma}^{(k)})^\top \boldsymbol{\Gamma}^{(k)}]\right)} \leq \hat{R}\sqrt{n_k m}. \tag{18}$$

Combining (17) and (18), we obtain

$$\mathbb{E}_\gamma[\sup_{\boldsymbol{W}} \langle \boldsymbol{W},\, \boldsymbol{X\Gamma} \rangle] = \mathbb{E}_\gamma \sup_{\boldsymbol{B},\boldsymbol{C}} \langle \boldsymbol{ABC}, \boldsymbol{X\Gamma} \rangle \leq \sum_{k=1}^K \sqrt{r_k} B_k \, \mathbb{E}_\gamma[\|\boldsymbol{X\Gamma}^{(k)}\|_F] \leq \sum_{k=1}^K \sqrt{r_k} B_k \hat{R}\sqrt{n_k m}. \tag{19}$$

**Algorithm 2** Estimation of $K$ and Subspace Segmentation [16]

**Input:** $\boldsymbol{V}_0$, $R$, $K_0$, and $\tau > 0$.

**Step 1**: Construct the affinity matrix $[\boldsymbol{F}]_{i,j} = ([\tilde{\boldsymbol{U}}\tilde{\boldsymbol{U}}^T]_{i,j})$, where $\tilde{\boldsymbol{U}}$ is formed by $[\boldsymbol{V}_0]_{:,1:R}$ with normalized rows;

**Step 2**: Estimate the number of subspaces $K$ by $K = \max(K_0, n - \text{int}\left(\sum_{i=1}^n f_\tau(\sigma_i(\boldsymbol{L}))\right)$ [16], where $\{\sigma_i(\boldsymbol{L})\}_{i=1}^n$ are the singular values of the Laplacian matrix $\boldsymbol{L} = \mathbf{I} - \boldsymbol{D}^{-\frac{1}{2}}\boldsymbol{F}\boldsymbol{D}^{-\frac{1}{2}}$, where $\boldsymbol{D} = \text{diag}\left(\sum_j[\boldsymbol{F}]_{1j}, \cdots, \sum_j[\boldsymbol{F}]_{nj}\right)$, $\text{int}(\cdot)$ is the function of the nearest integer, and $f_\tau(\sigma) = \begin{cases} 1, & \text{if } \sigma \geq \tau, \\ \log_2\left(1 + \frac{\sigma^2}{\tau^2}\right), & \text{otherwise.} \end{cases}$

**Step 3**: Construct an undirected graph by using the affinity matrix $\boldsymbol{F}$;

**Step 4**: Apply the NCut [30] to segment the vertices into $K$ clusters;

Substituting (19) into (15) leads to

$$\hat{G}_S(\mathcal{F}_{\text{AdaMSS}}) \leq \mathcal{L}(\phi)\frac{1}{m}\mathbb{E}_\gamma \sup_{\boldsymbol{W}}\langle \boldsymbol{W}, \boldsymbol{\Gamma}\boldsymbol{X}\rangle \leq \mathcal{L}(\phi)\sum_{k=1}^K \hat{R}B_k\sqrt{\frac{r_k n_k}{m}}.$$

$\square$

### B.2 Gaussian Complexity Analysis for LoRA and Its Single Subspace Variants

By the Vector-valued Gaussian complexity Generalization Bound Theorem [22], we can directly derive the Gaussian complexity of LoRA, as stated in Lemma 2.

**Lemma 2.** *[Gaussian complexity of the LoRA for a shallow Lipschitz neural network] For the class of spectrally bounded shallow Lipschitz network*

$$\mathcal{F}_{\text{lora}} = \{f_{\boldsymbol{W}}(\boldsymbol{x}) = \phi(\boldsymbol{x}\boldsymbol{W}) \mid \boldsymbol{W} = \boldsymbol{W}_0 + \Delta\boldsymbol{W}, \|\Delta\boldsymbol{W}\|_2 \leq B\} \tag{20}$$

*the Gaussian complexity of this function class is upper bounded as follows:*

$$\hat{G}_S(\mathcal{F}_{\text{lora}}) \leq \mathcal{L}(\phi)\hat{R}B\sqrt{\frac{rn}{m}}, \tag{21}$$

*where $r = \text{rank}(\Delta\boldsymbol{W})$, $n$ denotes the width of the weight matrix $\Delta\boldsymbol{W}$, $\mathcal{L}(\phi)$ is Lipschitz constant for function $\phi$, $\boldsymbol{X} = [\boldsymbol{x}_1^\top, \ldots, \boldsymbol{x}_m^\top] \in \mathbb{R}^{d\times m}$ for the samples $\{\boldsymbol{x}_i\}_{i=1}^m$, and $\max_{i=1,2,\cdots,m}\|\boldsymbol{x}_i\| \leq \hat{R}$.*

Similarly, by [22], we can obtain the same upper bound in (21) for the Gaussian complexity of PiSSA and other single subspace variants under the same assumptions.

## C Subspace Segmentation (Algorithm 2)

As Step 2 in Algorithm 2 incurs significant computational and memory costs, a practical alternative for large models is to fix $K = K_0$ for large models.

## D Parameter Count

Table 5: Comparison of parameter counts among different low-rank-based methods for $d = n$: $r$ represents the rank of the incremental update used in LoRA and PiSSA.

|  | LoRA | PiSSA | AdaMSS |
|---|---|---|---|
| # Trainable Parameters | $2rn$ | $2rn$ | $\sum_{k=1}^K(r_k n_k + r_k R_k)$ |

Given $\boldsymbol{W}^{(k)} = \boldsymbol{W}_0^{(k)} + \boldsymbol{A}^{(k)}\boldsymbol{B}^{(k)}\boldsymbol{C}^{(k)}$ for $k = 1, 2, \cdots, K$, where $\boldsymbol{B}^{(k)} \in \mathbb{R}^{R_k \times r_k}$ and $\boldsymbol{C}^{(k)} \in \mathbb{R}^{r_k \times n_k}$ are trainable, the total trainable parameter count in AdaMSS is given by $\sum_{k=1}^K(r_k n_k + r_k R_k)$,

which is always less than $2(\sum_{k=1}^{K} r_k)\max_{k=1,2,\cdots,K} n_k$ for $d = n$. Compared to the trainable parameter count in LoRA and PiSSA, as shown in Table 5, AdaMSS introduces significantly fewer parameters for $r = \sum_{k=1}^{K} r_k$, owing to the fact that $r_k \leq R_k \leq n_k$ and $n = \sum_{k=1}^{K} n_k$.

# E  Computation Complexity

By the proposed adaptation, we have $\boldsymbol{Y} = \boldsymbol{X}(\boldsymbol{W}_0 + \boldsymbol{ABC})$, where $\boldsymbol{X} \in \mathbb{R}^{\text{bach size} \times n}$ is the input. In table 6, we compare the computational complexity of gradient computation for LoRA, PiSSA, and AdaMSS. The table shows that AdaMSS's computational complexity is on the same order as LoRA and PiSSA when $\sum_{i=1}^{K} r_k = r$.

Table 6: Comparison of LoRA, PiSSA and AdaMSS for $d = n$, where underline denotes the trainable parameters, and $\boldsymbol{A} \in \mathbb{R}^{n\times r}$ and $\boldsymbol{B} \in \mathbb{R}^{r\times n}$ for LoRA and PiSSA.

|  | LoRA | PiSSA | AdaMSS |
|---|---|---|---|
| Forward | $\boldsymbol{Y} = \boldsymbol{X}\boldsymbol{W}_0 + \boldsymbol{X}\underline{\boldsymbol{AB}}$ | $\boldsymbol{Y} = \boldsymbol{X}\boldsymbol{W}_{res} + \boldsymbol{X}\underline{\boldsymbol{AB}}$ | $\boldsymbol{Y} = \boldsymbol{X}\boldsymbol{W}_0 + \boldsymbol{X}\boldsymbol{A}\underline{\boldsymbol{BC}}$ |
| Gradient | $\frac{\partial \mathcal{L}}{\partial \boldsymbol{A}} = \boldsymbol{X}^\top\left(\frac{\partial \mathcal{L}}{\partial \boldsymbol{Y}}\right)\boldsymbol{B}^\top$ | $\frac{\partial \mathcal{L}}{\partial \boldsymbol{A}} = \boldsymbol{X}^\top\left(\frac{\partial \mathcal{L}}{\partial \boldsymbol{Y}}\right)\boldsymbol{B}^\top$ | $\frac{\partial \mathcal{L}}{\partial \boldsymbol{B}} = (\boldsymbol{X}\boldsymbol{A})^\top\left(\frac{\partial \mathcal{L}}{\partial \boldsymbol{Y}}\right)\boldsymbol{C}^\top$ |
|  | $\frac{\partial \mathcal{L}}{\partial \boldsymbol{B}} = \boldsymbol{A}^\top\boldsymbol{X}^\top\left(\frac{\partial \mathcal{L}}{\partial \boldsymbol{Y}}\right)$ | $\frac{\partial \mathcal{L}}{\partial \boldsymbol{B}} = \boldsymbol{A}^\top\boldsymbol{X}^\top\left(\frac{\partial \mathcal{L}}{\partial \boldsymbol{Y}}\right)$ | $\frac{\partial \mathcal{L}}{\partial \boldsymbol{C}} = \boldsymbol{B}^\top(\boldsymbol{X}\boldsymbol{A})^\top\left(\frac{\partial \mathcal{L}}{\partial \boldsymbol{Y}}\right)$ |
| Cost | $\mathcal{O}(rn^2 + \text{bach size} \times n^2)$ | $\mathcal{O}(rn^2 + \text{bach size} \times n^2)$ | $\mathcal{O}(\sum_{i=1}^{K} r_k n^2 + \text{bach size} \times n^2)$ |

# F  Comparison of Optimizer Memory Efficiency and Training Speed

Since our method uses fewer trainable parameters, it exhibits superior memory efficiency, as shown in Tables 7 and 8.

Table 7: Optimizer memory consumption (MB) of different PEFT methods on `ViT-Large` using fp32 precision.

| Method | LoRA ($r = 16$) | PiSSA ($r = 8$) | LoRETTA ($r = 5$) | AdaMSS$_{\text{base}}$ ($r_k = 1$) |
|---|---|---|---|---|
| Memory (MB) | 18.84 | 9.6 | 1.584 | 2.136 |

Table 8: Optimizer memory consumption (MB) of different PEFT methods on `RoBERTa-Large` using fp32 precision.

| Method | LoRA ($r = 8$) | PiSSA ($r = 8$) | LoRETTA ($r = 5$) | AdaMSS ($r_k = 1$) |
|---|---|---|---|---|
| Memory (MB) | 9.6 | 9.6 | 1.58 | 0.54 |

Regarding training speed, as discussed in the main text, the computational complexity of our gradient updates remains on the same order as LoRA and PiSSA when $r = \sum_{k=1}^{K} r_k$, where $r$ is the rank hyperparameter used in LoRA and PiSSA. However, in practice, the multi-subspace adaptive budget allocation in our method accelerates training by selectively updating only the most important subspaces (see Table 9). Moreover, this adaptive budget allocation offers greater flexibility in balancing training efficiency and model performance. Specifically, the number of trainable subspaces is gradually reduced to a target value $K_{\text{target}}$ according to a smooth decay schedule with decay exponent $\rho = 3$. A larger $\rho$ results in a faster reduction in the number of trainable subspaces. Table 10 provides an ablation study demonstrating the impact of different $\rho$ values on both training speed and final performance.

Table 9: Average training time (in seconds) of different PEFT methods on `RoBERTa-Large` for natural language understanding task (STS-B).

| Method | LoRA ($r = 8$) | PiSSA ($r = 8$) | LoRETTA ($r = 5$) | AdaMSS ($r_k = 1, \sum_k r_k \geq 10$) |
|---|---|---|---|---|
| Avg. Time (s) | 5325.77 | 5174.04 | 5441.47 | 4972.03 |

Table 10: Training time (in seconds) and performance (PCC) of AdaMSS with varying $\rho$ on `RoBERTa-Large` for natural language understanding task (STS-B).

| $\rho$ | 5 | 10 | 15 |
|---|---|---|---|
| Avg. Time (s) | 4913.63 | 4780.97 | 4667.35 |
| PCC | $91.66_{\pm 0.03}$ | $91.52_{\pm 0.02}$ | $91.35_{\pm 0.03}$ |

# G    Ablation Study

## G.1    Comparison of Different Adaptive Budget Allocation Strategies

We compare our importance-score-based adaptive budget allocation strategy (described in the main text) with the following two alternative approaches:

- **Random adaptive budget allocation**: At each training step, $K_t$ subspaces are randomly selected, and only the corresponding parameters are updated.

- $\ell_1$**-norm-based adaptive budget allocation**: At each training step, the $\ell_1$-norm of the parameters associated with each subspace is computed, and the top-$K_t$ subspaces with the largest $\ell_1$-norms are selected for updating.

As shown in Table 11, both random and $\ell_1$-norm-based adaptive allocation strategies lead to considerable performance degradation. In contrast, our importance-score-based allocation method achieves the best performance for all cases.

Table 11: Performance comparison of different adaptive budget allocation strategies on the `ViT-Large` model.

| Methods \ Dataset | StanfordCars | CIFAR100 | FGVC |
|---|---|---|---|
| Importance-score-based adaptive budget allocation | $\mathbf{85.24}_{\pm 0.3}$ | $\mathbf{93.22}_{\pm 0.01}$ | $\mathbf{64.31}_{\pm 0.4}$ |
| Random adaptive budget allocation | $83.63_{\pm 0.30}$ | $93.11_{\pm 0.10}$ | $59.36_{\pm 0.72}$ |
| $\ell_1$-norm-based adaptive budget allocation | $79.89_{\pm 0.41}$ | $92.00_{\pm 0.21}$ | $40.77_{\pm 1.13}$ |

## G.2    Performance of AdaMSS With Varying $r_k$

This subsection explores how the performance of AdaMSS varies with different values of $r_k$ and $K_{\text{target}}$ using `ViT-Large`, where $r_k \in \{1, 2, 3\}$ and $K_{\text{target}} \in \{100, 200, 300, 400, 500\}$. Each configuration is evaluated on the StanfordCars, FGVC datasets, and CIFAR100. All results are presented in Figures 5. As shown in the results, increasing $r_k$ can lead to significant improvements in accuracy.

## G.3    Sensitivity Analysis of $\rho$ and $K_{\text{target}}$ in Adaptive Budget Allocation

We evaluate the sensitivity of AdaMSS to different values of $\rho \in \{1, 2, 3, 4, 5\}$ and $K_{\text{target}} \in \{100, 200, 300, 400\}$ using the `ViT-Large`, as shown in Tables 15–17. The results demonstrate that AdaMSS maintains robust performance across a wide range of $\rho$ and $K_{\text{target}}$ settings.

Table 12: Results of AdaMSS$_{\text{base}}$ under varying $\tau$ and $K_0$ values for StanfordCars.

| $\tau \backslash K_0$ | 1 | 5 | 10 | 15 | 20 |
|---|---|---|---|---|---|
| 0.001 | $84.25_{\pm 0.23}$ | $85.40_{\pm 0.19}$ | $85.40_{\pm 0.22}$ | $85.38_{\pm 0.25}$ | $85.67_{\pm 0.42}$ |
| 0.01 | $84.12_{\pm 0.24}$ | $85.20_{\pm 0.50}$ | $85.31_{\pm 0.27}$ | $85.41_{\pm 0.13}$ | $85.50_{\pm 0.21}$ |
| 0.05 | $84.77_{\pm 0.16}$ | $85.02_{\pm 0.26}$ | $85.23_{\pm 0.28}$ | $85.59_{\pm 0.27}$ | $85.70_{\pm 0.24}$ |
| 0.10 | $84.58_{\pm 0.29}$ | $84.93_{\pm 0.25}$ | $85.60_{\pm 0.26}$ | $85.36_{\pm 0.29}$ | $85.42_{\pm 0.23}$ |
| 0.15 | $84.72_{\pm 0.39}$ | $85.12_{\pm 0.23}$ | $85.38_{\pm 0.33}$ | $85.61_{\pm 0.20}$ | $85.63_{\pm 0.45}$ |
| 0.20 | $84.57_{\pm 0.09}$ | $84.86_{\pm 0.17}$ | $85.53_{\pm 0.21}$ | $85.39_{\pm 0.34}$ | $85.65_{\pm 0.19}$ |

Table 13: Results of AdaMSS$_{\text{base}}$ under varying $\tau$ and $K_0$ values for CIFAR100.

| $\tau \backslash K_0$ | 1 | 5 | 10 | 15 | 20 |
|---|---|---|---|---|---|
| 0.001 | $93.27_{\pm 0.09}$ | $93.44_{\pm 0.16}$ | $93.47_{\pm 0.06}$ | $93.57_{\pm 0.12}$ | $93.48_{\pm 0.15}$ |
| 0.01 | $93.38_{\pm 0.08}$ | $93.33_{\pm 0.10}$ | $93.50_{\pm 0.10}$ | $93.53_{\pm 0.15}$ | $93.57_{\pm 0.05}$ |
| 0.05 | $93.42_{\pm 0.12}$ | $93.43_{\pm 0.12}$ | $93.51_{\pm 0.09}$ | $93.45_{\pm 0.10}$ | $93.56_{\pm 0.09}$ |
| 0.10 | $93.40_{\pm 0.12}$ | $93.47_{\pm 0.09}$ | $93.58_{\pm 0.13}$ | $93.45_{\pm 0.12}$ | $93.64_{\pm 0.10}$ |
| 0.15 | $93.43_{\pm 0.12}$ | $93.43_{\pm 0.14}$ | $93.52_{\pm 0.06}$ | $93.55_{\pm 0.06}$ | $93.60_{\pm 0.12}$ |
| 0.20 | $93.43_{\pm 0.10}$ | $93.54_{\pm 0.05}$ | $93.48_{\pm 0.08}$ | $93.53_{\pm 0.04}$ | $93.51_{\pm 0.11}$ |

Table 14: Results of AdaMSS$_{\text{base}}$ under varying $\tau$ and $K_0$ values for FGVC.

| $\tau \backslash K_0$ | 1 | 5 | 10 | 15 | 20 |
|---|---|---|---|---|---|
| 0.001 | $61.36_{\pm 1.12}$ | $65.33_{\pm 0.55}$ | $66.11_{\pm 0.53}$ | $66.62_{\pm 0.78}$ | $66.98_{\pm 0.73}$ |
| 0.01 | $61.90_{\pm 0.55}$ | $64.58_{\pm 0.69}$ | $65.27_{\pm 0.64}$ | $66.10_{\pm 0.69}$ | $66.60_{\pm 0.68}$ |
| 0.05 | $61.64_{\pm 0.22}$ | $64.63_{\pm 0.98}$ | $65.51_{\pm 1.05}$ | $66.63_{\pm 0.48}$ | $67.44_{\pm 0.68}$ |
| 0.10 | $62.42_{\pm 0.33}$ | $64.49_{\pm 0.87}$ | $65.88_{\pm 0.68}$ | $66.29_{\pm 0.21}$ | $67.16_{\pm 0.82}$ |
| 0.15 | $62.56_{\pm 0.73}$ | $64.90_{\pm 1.02}$ | $65.72_{\pm 1.04}$ | $66.86_{\pm 0.30}$ | $66.44_{\pm 0.92}$ |
| 0.20 | $62.00_{\pm 0.51}$ | $64.76_{\pm 0.78}$ | $65.68_{\pm 0.93}$ | $66.65_{\pm 0.41}$ | $66.86_{\pm 0.60}$ |

Table 15: Results of AdaMSS under varying $\rho$ and $K_{\text{target}}$ values for StanfordCars.

| $\rho \backslash K_{\text{target}}$ | 100 | 200 | 300 | 400 | 500 |
|---|---|---|---|---|---|
| 1 | $84.78_{\pm 0.32}$ | $85.06_{\pm 0.16}$ | $85.34_{\pm 0.13}$ | $85.35_{\pm 0.11}$ | $85.16_{\pm 0.12}$ |
| 2 | $84.88_{\pm 0.36}$ | $85.11_{\pm 0.19}$ | $85.21_{\pm 0.33}$ | $85.26_{\pm 0.20}$ | $85.21_{\pm 0.41}$ |
| 3 | $85.03_{\pm 0.29}$ | $84.91_{\pm 0.24}$ | $85.21_{\pm 0.39}$ | $85.06_{\pm 0.20}$ | $85.02_{\pm 0.18}$ |
| 4 | $84.69_{\pm 0.20}$ | $84.95_{\pm 0.13}$ | $85.03_{\pm 0.29}$ | $85.02_{\pm 0.21}$ | $85.19_{\pm 0.13}$ |
| 5 | $84.89_{\pm 0.27}$ | $84.67_{\pm 0.25}$ | $85.04_{\pm 0.25}$ | $85.16_{\pm 0.26}$ | $85.37_{\pm 0.36}$ |

Table 16: Results of AdaMSS under varying $\rho$ and $K_{\text{target}}$ values for CIFAR100.

| $\rho \backslash K_{\text{target}}$ | 100 | 200 | 300 | 400 | 500 |
|---|---|---|---|---|---|
| 1 | $93.47_{\pm 0.10}$ | $93.49_{\pm 0.11}$ | $93.41_{\pm 0.14}$ | $93.56_{\pm 0.10}$ | $93.34_{\pm 0.12}$ |
| 2 | $93.41_{\pm 0.14}$ | $93.44_{\pm 0.07}$ | $93.54_{\pm 0.08}$ | $93.42_{\pm 0.13}$ | $93.47_{\pm 0.07}$ |
| 3 | $93.42_{\pm 0.14}$ | $93.38_{\pm 0.05}$ | $93.56_{\pm 0.09}$ | $93.46_{\pm 0.14}$ | $93.55_{\pm 0.08}$ |
| 4 | $93.25_{\pm 0.11}$ | $93.53_{\pm 0.09}$ | $93.40_{\pm 0.10}$ | $93.48_{\pm 0.15}$ | $93.50_{\pm 0.07}$ |
| 5 | $93.35_{\pm 0.15}$ | $93.45_{\pm 0.11}$ | $93.37_{\pm 0.09}$ | $93.41_{\pm 0.12}$ | $93.41_{\pm 0.12}$ |

Table 17: Results of AdaMSS under varying $\rho$ and $K_{\text{target}}$ values for FGVC.

| $\rho \backslash K_{\text{target}}$ | 100 | 200 | 300 | 400 | 500 |
|---|---|---|---|---|---|
| 1 | $64.76_{\pm 0.55}$ | $65.56_{\pm 0.60}$ | $64.98_{\pm 0.81}$ | $64.90_{\pm 0.98}$ | $65.73_{\pm 0.48}$ |
| 2 | $64.55_{\pm 0.73}$ | $64.38_{\pm 0.83}$ | $65.14_{\pm 0.89}$ | $65.11_{\pm 0.57}$ | $64.99_{\pm 0.70}$ |
| 3 | $63.73_{\pm 0.31}$ | $64.16_{\pm 1.22}$ | $64.77_{\pm 0.34}$ | $65.33_{\pm 0.64}$ | $65.77_{\pm 0.21}$ |
| 4 | $63.02_{\pm 0.89}$ | $64.35_{\pm 0.62}$ | $64.64_{\pm 0.81}$ | $64.54_{\pm 0.47}$ | $65.13_{\pm 0.61}$ |
| 5 | $62.74_{\pm 1.53}$ | $63.43_{\pm 0.63}$ | $65.12_{\pm 0.83}$ | $64.28_{\pm 0.99}$ | $65.32_{\pm 0.73}$ |

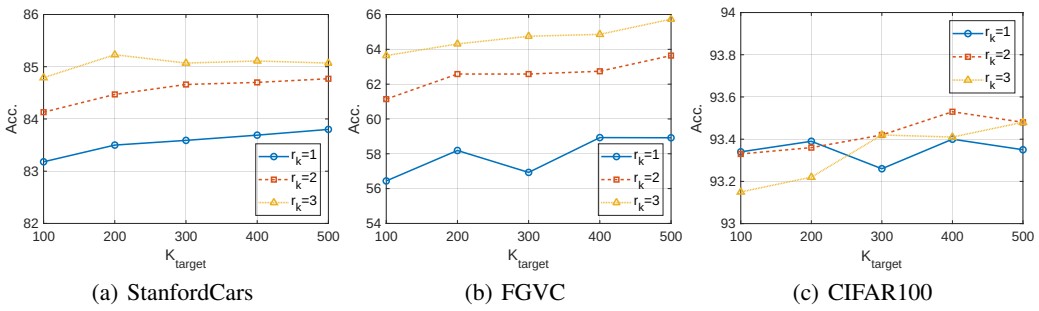

(a) StanfordCars      (b) FGVC      (c) CIFAR100

Figure 5: The performance of AdaMSS with different values of $r_k$ and $K_{\text{target}}$.

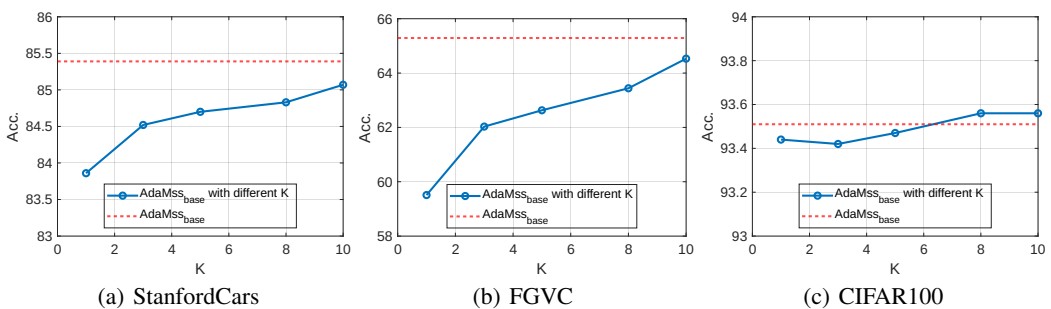

(a) StanfordCars      (b) FGVC      (c) CIFAR100

Figure 6: The performance of AdaMSS$_{\text{base}}$ with different values of $K$ for $r_k = 3$.

### G.4 Sensitivity Analysis of the Subspace Number $K$

#### G.4.1 $K$ estimated by Algorithm 2 for given $\tau$ and $K_0$

In Algorithm 2, the threshold $\tau$ is used to detect near-zero singular values in the normalized Laplacian matrix and is set to a small value. The parameter $K_0$ serves as a lower bound on the estimated number of subspaces. We evaluate different settings of $\tau \in \{0.001, 0.01, 0.05, 0.10, 0.15, 0.20\}$ and $K_0 \in \{1, 5, 10, 15, 20\}$ using `ViT-Large` with AdaMSS$_{\text{base}}$, as reported in Tables 12–14. The results demonstrate that AdaMSS$_{\text{base}}$ maintains robust performance across all three datasets (StanfordCars, CIFAR100, and FGVC) when $K_0 \geq 10$, regardless of the specific choice of $\tau$.

#### G.4.2 $K$ given manually.

This subsection investigate the effect of varying $K$ on the performance of AdaMSS$_{\text{base}}$ at $r_k = 3$, comparing estimated $K$ (shown as red dashed lines) with given values of $K$. As shown in Figures 6, increasing $K$ can also lead to significant improvements in accuracy. Notably, the improvement associated with increasing $K$ highlights the advantage of the multi-subspace approach.

### G.5 Initialization of AdaMSS

In this subsection, we compare two initialization strategies for $\boldsymbol{B}^{(k)}$: the proposed orthogonal initialization (as described in Steps 6-8 of Algorithm 1) and a random initialization following the LoRA strategy. While the latter avoids the computations in Steps 6-7 and simplifies the initialization procedure, we do not recommend this approach for AdaMSS. This is because randomly initializing $\boldsymbol{B}^{(k)}$ disrupts the preservation of structural information contained in $\hat{\boldsymbol{W}}_0^{(k)}$. As reparameterization of (8), the decomposition $\boldsymbol{A}^{(k)}\boldsymbol{B}^{(k)}$ in (10) is designed to remain as close as possible to the subspace spanned by $\hat{\boldsymbol{W}}_0^{(k)}$, which is crucial for AdaMSS—a method derived from a subspace segmentation perspective.

Figure 7 analyzes the impact of the two initialization strategies on `ViT-Large` using the StanfordCars and FGVC datasets, under varying values of $K$ and $r_k$. The results show that orthogonal initialization

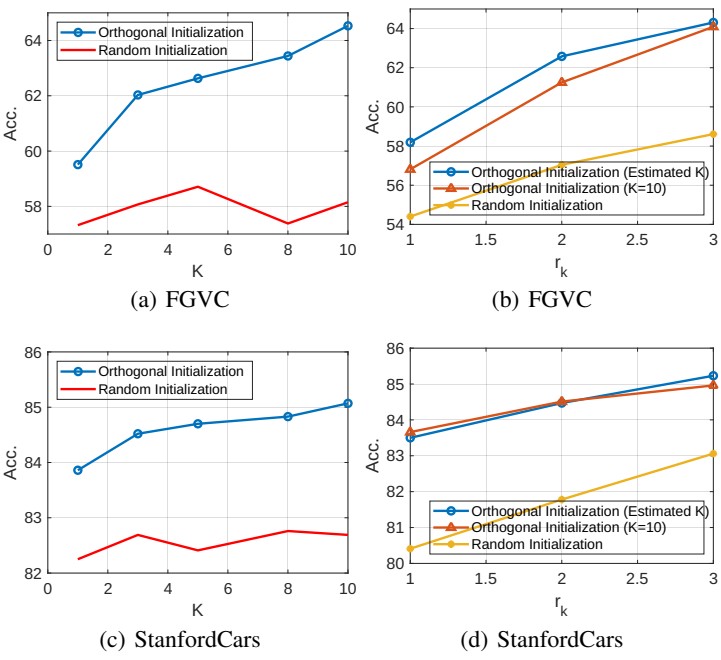

Figure 7: The performance of AdaMSS with different initialization strategies.

consistently yields superior performance across all configurations compared to random initialization, highlighting the effectiveness of orthogonal initialization in preserving subspace structure.

## H  Experimental Setup and Hyper-parameter Configuration

For a fair comparison, we replicate the experimental setups in [1, 2, 24]:

- IC and NLU: The reported results are averaged over five random seeds. All models are fine-tuned by updating only the query and value projection matrices, while the classification head is updated for every method.

- NLG: Following [2], we use the datasets listed in Table 21, and all experiments are conducted on 100K-example subsets and trained for a single epoch. The results are averaged over three runs.

- A detailed list of all hyper-parameters and settings can be found in the Tables 18-20.

In our experiments, we set $R = 100$, $\rho = 3$, and $\tau = 0.01$. If not specified, $K_0$ is set to 10.

Table 18: Hyper-Parameter Configuration for AdaMSS$_{\text{base}}$ and AdaMSS on IC.

| Model | Hyperparameter | OxfordPets | StanfordCars | CIFAR10 | EuroSAT | FGVC | RESISC45 | CIFAR100 |
|---|---|---|---|---|---|---|---|---|
| Both | Optimizer | | | | AdamW | | | |
| | Batch Size | | | | 10 | | | |
| | Epochs | | | | 10 | | | |
| | Seeds | | | {7,77,777,7777,77777} | | | | |
| ViT-Base | Learning Rate | 0.005 | 0.01 | 0.01 | 0.01 | 0.01 | 0.01 | 0.01 |
| | Learning Rate (Head) | 0.005 | 0.005 | 0.005 | 0.0005 | 0.005 | 0.005 | 0.005 |
| | Weight Decay | 0.0005 | 0.0 | 0.05 | 0.05 | 0.0005 | 0.0005 | 0.05 |
| ViT-Large | Learning Rate | 0.001 | 0.01 | 0.01 | 0.01 | 0.01 | 0.01 | 0.01 |
| | Learning Rate (Head) | 0.0005 | 0.005 | 0.05 | 0.0005 | 0.0005 | 0.0005 | 0.05 |
| | Weight Decay | 0.0005 | 0.1 | 0.1 | 0.01 | 0.0005 | 0.1 | 0.05 |

Table 19: Hyper-Parameter Configuration for AdaMSS$_{\text{base}}$ and AdaMSS on NLU.

| Model | Hyperparameter | SST-2 | MRPC | CoLA | QNLI | RTE | STS-B |
|---|---|---|---|---|---|---|---|
| Both | Optimizer | AdamW | | | | | |
| | Batch Size | 32 | | | | | |
| | Epochs | 100 | | | | | |
| | Seeds | {0,11111,22222,33333,444444} | | | | | |
| RoBERTa-Base | Learning Rate | 0.001 | 0.01 | 0.001 | 0.001 | 0.0005 | 0.001 |
| | Learning Rate (Head) | 0.005 | 0.0005 | 0.005 | 0.005 | 0.005 | 0.005 |
| | Weight Decay | 0.0005 | 0.0 | 0.005 | 0.005 | 0.005 | 0.005 |
| RoBERTa-Large | Learning Rate | 0.001 | 0.001 | 0.005 | 0.0005 | 0.005 | 0.001 |
| | Learning Rate (Head) | 0.0005 | 5e-05 | 0.0005 | 0.05 | 0.005 | 0.0005 |
| | Weight Decay | 0.0 | 0.005 | 0.1 | 0.005 | 0.5 | 0.0005 |

Table 20: Hyper-Parameter Configuration for AdaMSS$_{\text{base}}$ and AdaMSS on NLG.

| Model | Hyperparameter | GSM8K | MATH |
|---|---|---|---|
| Both | Optimizer | AdamW | |
| | Batch Size | 4 | |
| | Epochs | 1 | |
| LLaMA 2-7B, Mistral-7B | Learning Rate | 6e-4 | |
| Gemma-7B | Learning Rate | 2e-4 | |

# I  More Evidence for Multiple Subspaces Structure in Pretrained Network Weights

In this section, we provide additional numerical evidence for the presence of multi-subspace structures in pretrained models across layers and tasks.

Beyond the approximate block-diagonal patterns shown in Figure 2, we also examine the distribution of weight column vectors through the singular values of a Laplacian matrix constructed from the principal components. Tables 22–25 report the singular value distributions of these Laplacian matrices, computed from the principal components of the query weight matrices at each layer of several pretrained models, including ViT-Large, LLaMA 2-7B, Mistral-7B, and Gemma-7B. The results show that, for most layers, the dominant singular values remain substantially large, while the trailing ones are close to zero (e.g., below 0.01).

The number of near-zero singular values can be interpreted as an estimate of the number of disjoint subspaces in the weight space. This offers further numerical evidence for the existence of multi-subspace structures within the principal components of pretrained weights.

# J  Related Works

## J.1  Parameter-Efficient Fine-Tuning (PEFT)

In recent years, a variety of PEFT methods have been proposed to adapt large pre-trained models to downstream tasks while minimizing the number of trainable parameters. Broadly, current PEFT methods include prefix-tuning [31, 32], adapter-based methods [33], sparse fine-tuning [34, 24], orthogonal fine-tuning [35, 36], and low-rank-based adaptation [1, 2, 3, 4, 5, 6, 7, 8, 9]. Specifically, prefix-tuning methods introduce a small task-specific trainable vectors, known as prefixes, while keeping the original model parameters frozen [31, 32]. In contrast, adapter-based methods [33] insert small trainable modules within each layer of the model, allowing efficient task adaptation by updating only these modules. Meanwhile, sparse fine-tuning, such as BitFit [34] and FourierFT [24], update only a small subset of parameters in bias terms or transformed weights. On another front, orthogonal fine-tuning introduces a learnable orthogonal transformation applied to the pre-trained weights, with the goal of preserving the model's original capabilities during adaptation [35, 36]. In addition to the above, low-rank-based adaptation methods, for example LoRA [1], PiSSA [2], LoRA-GA [5], Foura

Table 21: The experimental setup

| Fine-tuned on | Evaluated Datasets |
|---|---|
| MetaMathQA [28] | GSM8K [13], MATH [28] |

Table 22: Singular value distribution of the Laplacian matrices constructed from the principal components of the query weight matrices at each layer of the pretrained `ViT-Large` model.

| Layer | $\sigma_1$ | $\sigma_{101}$ | $\sigma_{201}$ | $\sigma_{301}$ | $\sigma_{401}$ | $\sigma_{501}$ | $\sigma_{601}$ | $\sigma_{701}$ | $\sigma_{801}$ | $\sigma_{901}$ | $\sigma_{1001}$ | $\sigma_{1024}$ |
|---|---|---|---|---|---|---|---|---|---|---|---|---|
| 1 | 1.00 | 0.87 | 0.78 | 0.59 | 0.38 | 0.21 | 0.09 | 0.02 | 0.00 | 0.00 | 0.00 | 0.00 |
| 3 | 1.00 | 0.83 | 0.78 | 0.72 | 0.65 | 0.57 | 0.47 | 0.37 | 0.24 | 0.10 | 0.00 | 0.00 |
| 5 | 1.00 | 0.83 | 0.81 | 0.79 | 0.77 | 0.74 | 0.70 | 0.64 | 0.54 | 0.38 | 0.05 | 0.00 |
| 7 | 1.00 | 0.82 | 0.81 | 0.79 | 0.77 | 0.74 | 0.71 | 0.67 | 0.60 | 0.47 | 0.01 | 0.00 |
| 9 | 1.00 | 0.82 | 0.81 | 0.79 | 0.78 | 0.76 | 0.74 | 0.71 | 0.67 | 0.56 | 0.00 | 0.00 |
| 11 | 1.00 | 0.83 | 0.81 | 0.80 | 0.79 | 0.77 | 0.75 | 0.73 | 0.68 | 0.55 | 0.00 | 0.00 |
| 13 | 1.00 | 0.83 | 0.81 | 0.80 | 0.78 | 0.76 | 0.74 | 0.71 | 0.66 | 0.52 | 0.00 | 0.00 |
| 15 | 0.99 | 0.81 | 0.79 | 0.77 | 0.76 | 0.73 | 0.71 | 0.67 | 0.63 | 0.53 | 0.00 | 0.00 |
| 17 | 1.00 | 0.82 | 0.80 | 0.79 | 0.77 | 0.76 | 0.74 | 0.71 | 0.68 | 0.60 | 0.00 | 0.00 |
| 19 | 1.00 | 0.82 | 0.81 | 0.80 | 0.78 | 0.77 | 0.76 | 0.74 | 0.71 | 0.66 | 0.00 | 0.00 |
| 21 | 1.00 | 0.82 | 0.81 | 0.80 | 0.80 | 0.79 | 0.78 | 0.77 | 0.75 | 0.72 | 0.00 | 0.00 |
| 23 | 1.00 | 0.83 | 0.82 | 0.81 | 0.80 | 0.80 | 0.79 | 0.78 | 0.77 | 0.75 | 0.12 | 0.00 |

Table 23: Singular value distribution of the Laplacian matrices constructed from the principal components of the query weight matrices at each layer of the pretrained `LLaMA 2-7B` model.

| Layer | $\sigma_1$ | $\sigma_{501}$ | $\sigma_{1001}$ | $\sigma_{1501}$ | $\sigma_{2001}$ | $\sigma_{2501}$ | $\sigma_{3001}$ | $\sigma_{3501}$ | $\sigma_{4001}$ | $\sigma_{4092}$ | $\sigma_{4093}$ | $\sigma_{4094}$ | $\sigma_{4095}$ | $\sigma_{4096}$ |
|---|---|---|---|---|---|---|---|---|---|---|---|---|---|---|
| 1 | 1.00 | 0.31 | 0.19 | 0.13 | 0.08 | 0.04 | 0.02 | 0.00 | 0.00 | 0.00 | 0.00 | 0.00 | 0.00 | 0.00 |
| 3 | 1.00 | 0.51 | 0.48 | 0.43 | 0.36 | 0.27 | 0.17 | 0.08 | 0.01 | 0.00 | 0.00 | 0.00 | 0.00 | 0.00 |
| 5 | 1.00 | 0.50 | 0.47 | 0.45 | 0.42 | 0.38 | 0.31 | 0.19 | 0.03 | 0.00 | 0.00 | 0.00 | 0.00 | 0.00 |
| 7 | 0.99 | 0.46 | 0.41 | 0.37 | 0.34 | 0.30 | 0.24 | 0.15 | 0.03 | 0.00 | 0.00 | 0.00 | 0.00 | 0.00 |
| 9 | 1.00 | 0.43 | 0.38 | 0.35 | 0.32 | 0.28 | 0.22 | 0.13 | 0.02 | 0.00 | 0.00 | 0.00 | 0.00 | 0.00 |
| 11 | 1.00 | 0.50 | 0.46 | 0.43 | 0.38 | 0.31 | 0.24 | 0.15 | 0.03 | 0.01 | 0.00 | 0.00 | 0.00 | 0.00 |
| 13 | 1.00 | 0.47 | 0.43 | 0.40 | 0.36 | 0.31 | 0.24 | 0.17 | 0.05 | 0.01 | 0.01 | 0.01 | 0.01 | 0.00 |
| 15 | 1.00 | 0.50 | 0.44 | 0.38 | 0.32 | 0.26 | 0.20 | 0.13 | 0.03 | 0.01 | 0.01 | 0.01 | 0.00 | 0.00 |
| 17 | 1.00 | 0.49 | 0.45 | 0.41 | 0.37 | 0.32 | 0.26 | 0.17 | 0.03 | 0.00 | 0.00 | 0.00 | 0.00 | 0.00 |
| 19 | 1.00 | 0.48 | 0.43 | 0.40 | 0.37 | 0.33 | 0.30 | 0.22 | 0.06 | 0.01 | 0.01 | 0.01 | 0.01 | 0.01 |
| 21 | 1.00 | 0.47 | 0.40 | 0.35 | 0.32 | 0.28 | 0.23 | 0.15 | 0.03 | 0.00 | 0.00 | 0.00 | 0.00 | 0.00 |
| 23 | 1.00 | 0.44 | 0.38 | 0.35 | 0.33 | 0.30 | 0.27 | 0.18 | 0.02 | 0.00 | 0.00 | 0.00 | 0.00 | 0.00 |
| 25 | 1.00 | 0.46 | 0.38 | 0.35 | 0.32 | 0.29 | 0.26 | 0.15 | 0.01 | 0.00 | 0.00 | 0.00 | 0.00 | 0.00 |
| 27 | 1.00 | 0.46 | 0.41 | 0.39 | 0.36 | 0.34 | 0.30 | 0.19 | 0.03 | 0.00 | 0.00 | 0.00 | 0.00 | 0.00 |
| 29 | 1.00 | 0.47 | 0.45 | 0.43 | 0.41 | 0.39 | 0.36 | 0.31 | 0.09 | 0.01 | 0.01 | 0.01 | 0.01 | 0.00 |
| 31 | 1.00 | 0.48 | 0.45 | 0.43 | 0.42 | 0.40 | 0.37 | 0.30 | 0.10 | 0.02 | 0.02 | 0.01 | 0.01 | 0.01 |

[8], and RoSA [7], approximate weight updates using low-rank matrices, significantly reducing the number of trainable parameters while maintaining performance. These techniques provide effective strategies for adapting large models to new tasks with less computational overhead.

## J.2 Low-Rank-Based Adaptation Methods

Inspired by the success of LoRA [1], a large number of low-rank-based adaptation methods have emerged over the past three years [2, 3, 4, 5, 6, 7, 8, 9, 10, 11]. LoRA assumes a fixed rank for all incremental matrices across different layers, thereby ignoring the diverse importance of different weight parameters. To address this limitation, AdaLoRA [3] was proposed to allocates parameter budgets by adaptively adjusting the rank during training. Following AdaLoRA, research attention has shifted toward improving the initialization strategies of $A$ and $B$ in LoRA, as opposed to relying on random initialization. Representative works include PiSSA [2] and LoRA-GA[5]. In addition, by incorporating both low-rankness and sparsity constraints, RoSA [7] enhances low-rank adaptation to enable more efficient parameter utilization. In another line of work, FouRA [8] extends LoRA into the frequency domain, yielding disentangled feature spaces that enable fine-grained control and editing. Motivated by the success of tensor decomposition in data compression [37], LoTR [11] and LoRETTA [10] leverage Tucker decomposition and Tensor-train decomposition, respectively, to achieve more compact representations of the low-rank update matrices.

Table 24: Singular value distribution of the Laplacian matrices constructed from the principal components of the query weight matrices at each layer of the pretrained `Mistral-7B` model.

| Layer | $\sigma_1$ | $\sigma_{501}$ | $\sigma_{1001}$ | $\sigma_{1501}$ | $\sigma_{2001}$ | $\sigma_{2501}$ | $\sigma_{3001}$ | $\sigma_{3501}$ | $\sigma_{4001}$ | $\sigma_{4092}$ | $\sigma_{4093}$ | $\sigma_{4094}$ | $\sigma_{4095}$ | $\sigma_{4096}$ |
|---|---|---|---|---|---|---|---|---|---|---|---|---|---|---|
| 1 | 1.00 | 0.29 | 0.15 | 0.09 | 0.05 | 0.02 | 0.01 | 0.00 | 0.00 | 0.00 | 0.00 | 0.00 | 0.00 | 0.00 |
| 3 | 1.00 | 0.49 | 0.46 | 0.43 | 0.41 | 0.38 | 0.28 | 0.16 | 0.03 | 0.00 | 0.00 | 0.00 | 0.00 | 0.00 |
| 5 | 1.00 | 0.48 | 0.44 | 0.40 | 0.36 | 0.32 | 0.25 | 0.17 | 0.06 | 0.01 | 0.01 | 0.01 | 0.01 | 0.00 |
| 7 | 1.00 | 0.47 | 0.43 | 0.39 | 0.35 | 0.31 | 0.23 | 0.14 | 0.04 | 0.01 | 0.01 | 0.00 | 0.00 | 0.00 |
| 9 | 1.00 | 0.48 | 0.44 | 0.41 | 0.38 | 0.34 | 0.27 | 0.19 | 0.06 | 0.01 | 0.01 | 0.00 | 0.00 | 0.00 |
| 11 | 1.00 | 0.47 | 0.42 | 0.37 | 0.32 | 0.27 | 0.21 | 0.11 | 0.02 | 0.00 | 0.00 | 0.00 | 0.00 | 0.00 |
| 13 | 0.99 | 0.42 | 0.33 | 0.27 | 0.21 | 0.16 | 0.11 | 0.06 | 0.01 | 0.00 | 0.00 | 0.00 | 0.00 | 0.00 |
| 15 | 1.00 | 0.45 | 0.38 | 0.33 | 0.27 | 0.22 | 0.16 | 0.11 | 0.03 | 0.01 | 0.01 | 0.01 | 0.00 | 0.00 |
| 17 | 1.00 | 0.48 | 0.41 | 0.35 | 0.30 | 0.24 | 0.19 | 0.12 | 0.03 | 0.01 | 0.01 | 0.01 | 0.01 | 0.01 |
| 19 | 1.00 | 0.40 | 0.32 | 0.27 | 0.23 | 0.18 | 0.14 | 0.08 | 0.02 | 0.00 | 0.00 | 0.00 | 0.00 | 0.00 |
| 21 | 1.00 | 0.47 | 0.41 | 0.36 | 0.32 | 0.27 | 0.21 | 0.15 | 0.04 | 0.01 | 0.01 | 0.00 | 0.00 | 0.00 |
| 23 | 1.00 | 0.46 | 0.42 | 0.39 | 0.36 | 0.33 | 0.27 | 0.16 | 0.02 | 0.00 | 0.00 | 0.00 | 0.00 | 0.00 |
| 25 | 1.00 | 0.44 | 0.40 | 0.37 | 0.34 | 0.31 | 0.25 | 0.18 | 0.06 | 0.01 | 0.01 | 0.00 | 0.00 | 0.00 |
| 27 | 1.00 | 0.47 | 0.43 | 0.41 | 0.38 | 0.34 | 0.30 | 0.22 | 0.07 | 0.01 | 0.01 | 0.01 | 0.01 | 0.01 |
| 29 | 1.00 | 0.45 | 0.41 | 0.38 | 0.35 | 0.32 | 0.26 | 0.17 | 0.03 | 0.00 | 0.00 | 0.00 | 0.00 | 0.00 |
| 31 | 1.00 | 0.46 | 0.42 | 0.39 | 0.36 | 0.32 | 0.26 | 0.16 | 0.02 | 0.00 | 0.00 | 0.00 | 0.00 | 0.00 |

Table 25: Singular value distribution of the Laplacian matrices constructed from the principal components of the query weight matrices at each layer of the pretrained `Gemma-7B` model.

| Layer | $\sigma_1$ | $\sigma_{501}$ | $\sigma_{1001}$ | $\sigma_{1501}$ | $\sigma_{2001}$ | $\sigma_{2501}$ | $\sigma_{3001}$ | $\sigma_{3501}$ | $\sigma_{4001}$ | $\sigma_{4092}$ | $\sigma_{4093}$ | $\sigma_{4094}$ | $\sigma_{4095}$ | $\sigma_{4096}$ |
|---|---|---|---|---|---|---|---|---|---|---|---|---|---|---|
| 1 | 1.00 | 0.50 | 0.46 | 0.44 | 0.41 | 0.37 | 0.29 | 0.15 | 0.00 | 0.00 | 0.00 | 0.00 | 0.00 | 0.00 |
| 3 | 1.00 | 0.49 | 0.46 | 0.45 | 0.43 | 0.41 | 0.39 | 0.32 | 0.12 | 0.01 | 0.01 | 0.01 | 0.01 | 0.00 |
| 5 | 1.00 | 0.48 | 0.45 | 0.43 | 0.41 | 0.39 | 0.36 | 0.26 | 0.03 | 0.00 | 0.00 | 0.00 | 0.00 | 0.00 |
| 7 | 1.00 | 0.49 | 0.47 | 0.45 | 0.44 | 0.42 | 0.38 | 0.28 | 0.08 | 0.00 | 0.00 | 0.00 | 0.00 | 0.00 |
| 9 | 1.00 | 0.50 | 0.47 | 0.44 | 0.41 | 0.37 | 0.33 | 0.24 | 0.07 | 0.00 | 0.00 | 0.00 | 0.00 | 0.00 |
| 11 | 1.00 | 0.48 | 0.45 | 0.43 | 0.42 | 0.39 | 0.35 | 0.25 | 0.05 | 0.00 | 0.00 | 0.00 | 0.00 | 0.00 |
| 13 | 1.00 | 0.50 | 0.47 | 0.44 | 0.42 | 0.37 | 0.29 | 0.20 | 0.07 | 0.00 | 0.00 | 0.00 | 0.00 | 0.00 |
| 15 | 1.00 | 0.49 | 0.44 | 0.38 | 0.29 | 0.18 | 0.10 | 0.04 | 0.00 | 0.00 | 0.00 | 0.00 | 0.00 | 0.00 |
| 17 | 1.00 | 0.50 | 0.45 | 0.41 | 0.36 | 0.30 | 0.23 | 0.14 | 0.04 | 0.01 | 0.00 | 0.00 | 0.00 | 0.00 |
| 19 | 0.99 | 0.52 | 0.45 | 0.38 | 0.30 | 0.22 | 0.15 | 0.08 | 0.01 | 0.00 | 0.00 | 0.00 | 0.00 | 0.00 |
| 21 | 0.99 | 0.48 | 0.43 | 0.39 | 0.34 | 0.27 | 0.19 | 0.11 | 0.01 | 0.00 | 0.00 | 0.00 | 0.00 | 0.00 |
| 23 | 1.00 | 0.51 | 0.42 | 0.36 | 0.31 | 0.25 | 0.19 | 0.11 | 0.02 | 0.00 | 0.00 | 0.00 | 0.00 | 0.00 |
| 25 | 1.00 | 0.49 | 0.41 | 0.33 | 0.27 | 0.20 | 0.11 | 0.04 | 0.00 | 0.00 | 0.00 | 0.00 | 0.00 | 0.00 |
| 27 | 1.00 | 0.46 | 0.43 | 0.40 | 0.37 | 0.33 | 0.28 | 0.19 | 0.05 | 0.01 | 0.01 | 0.00 | 0.00 | 0.00 |

