# OpenReview forum: "AdaMSS: Adaptive Multi-Subspace Approach for Parameter-Efficient Fine-Tuning"
_NeurIPS.cc/2025/Conference — NeurIPS 2025 poster_

### Official Review · Reviewer_i4NE · 2025-06-30

**Clarity:** 3
**Significance:** 3
**Originality:** 3
**Rating:** 4
**Confidence:** 4

**Summary:**

- AdaMSS proposes a new parametrization for low-rank adaptation based on subspace ssegmentation.
- The weights in transformer models can be written as $\hat{W}_0 = \hat{W}_0 Z_0$, with $\hat{W}_0$ being the rank-truncated SVD of the pretrained weights $W_0$, and $Z_0$ has an approximate block diagonal structure alon with an analytical solution.
- This formulation is an application of the Low-Rank Representation (LRR) model.
- The row space is segmented into $K$ smaller subspaces (blocks in the block diagonal matrix $Z_0$), where each block contains columns of $W_0$ assined to the block.
- Based on this formulation, the weights can be written as $W_0^{(k)} = \hat{W}_0^{(k)}(Z_0^*)^{(k)} + \bar{W}_{res}^{(k)}$
- The paper proposes a residual update $\Delta Z^{(k)}$, so that the final low-rank update can be written as a decomposition $W^{(k)} = W_0^{(k)} + A^{(k)}B^{(k)}C^{(k)}$, with A and B initialized using the SVD of $W_0$ and C initialized using 0.
- The paper further proposes to freeze ranks progressively based on the gradients of $B$ and $C$, leading to adaptive budet allocation.
- AdaMSS achieves a lower theoretical bound on generalization.

**Questions:**

For all the experiments, which modules are fine-tuned (for e.g, Query Value, etc.)?

**Ethical Concerns:**

["NO or VERY MINOR ethics concerns only"]

**Final Justification:**

My main concern was regarding the overhead of the method compared to LoRA and PiSSA, and whether it is justified. Because of the following two points, I have increased my score and recommend acceptance:

- The authors have demonstrated that with AdaMSS can adapt models to downstream tasks better than prior methods with comparable number of parameters. This shows the presence of meaningful subspaces in pretrained model weights, which can potentially open more avenues for future research.
- AdaMSS shows theoretical guarantees for loer test error.

**Limitations:**

Please refer to point (1) in the weaknesses for a limitation that I have observed.

**Quality:**

3

**Strengths And Weaknesses:**

## Strengths
- The AdaMSS formulation enables fine-tuning with very few additional parameters
- The update is theoretically motivated from subspace segmentation, and the verification experiments confirm the theory
- The bound on generalization is lower than prior methods, potentially offering better generalization capabilities

## Weaknesses
1. For the amount of complexity added, the performance improvement over prior methods like PiSSA are small (especially since PiSSA is much simpler)
    - The initialization of the A and B requires computing the SVD of weight matrices, which can be quite costly for large models
    - The authors mention that the matrices can be randomly initialized, no empirical experiments have been performed
    - While this method does reduce the number of parameters, in practice, rank 16 is very economical, and anything below that does not really reduce the training time/memory requirements much especially for large models.
    - The question then is, what other benefits/insights does the method offer? Can the authors perform some experiments that demonstrate generalization?

2. For the experiments on large language models, the comparison with equivalent trainable parameter counts for LoRA and PiSSA is missing
    - A comparison with equivalent parameter count would be useful to see how quickly any method can adapt with minimal parameters, showing more benefit of the structured adaptation

3. Missing citation and comparison
    - The residual update formulation in equation 8 ($W^{(k)} = W_0^{(k)} + \hat{W}_0^{(k)}\Delta Z^{(k)}$) is very similar to [1]. Can the authors explain the difference and add a comparison?

Overall, the idea is interesting and the evaluation is good, and I would consider increasing my score if the authors can address my concerns.

---

## References

[1] Chinmay Savadikar, Xi Song, and Tianfu Wu. "Generative Parameter-Efficient Fine-Tuning." arXiv preprint arXiv:2312.00700 (2023).

---

> ### Author Rebuttal · Authors · 2025-07-31
>
> We sincerely appreciate the reviewer’s recognition of our work and the constructive feedback and insightful questions provided. Below, we address the concerns and questions raised.
>
> ---
> **R1. Performance improvement over prior methods like PiSSA**
> Thank you so much for your comments. Our method consistently outperforms other approaches—including **PiSSA**—in terms of accuracy, often with **comparable or significantly fewer trainable parameters**.
> - For **Image Classification (IC)**, our method achieves over **6% higher accuracy** than PiSSA on **ViT-Base** with a **similar parameter budget**.
> - For **Natural Language Understanding (NLU)**, our method achieves over **7% higher accuracy** than PiSSA on **RoBERTa-Large**, while using much fewer parameters.
>   In our newly added experiments (as suggested by the reviewer), we have included results for **PiSSA with $r=1$** (see Tables 1–2).  These results show that, under similar parameter conditions, our method performs  **consistently** across tasks on both **RoBERTa-Large** and **RoBERTa-Base**, and achieves **substantially higher average performance** compared to PiSSA.
> - For **Natural Language Generation (NLG)** (see additional results in the supplementary material), our method outperforms PiSSA on **LLaMA 2-7B** by: **7% higher accuracy** on **GSM8K**, and **1.7% higher accuracy** on **MATH**, while using **fewer trainable parameters**.
>
> Below we summarize tasks where our method achieves **at least 3% higher accuracy** on average compared to other methods. **Bold entries** indicate cases where AdaMSS **reduces the number of trainable parameters by more than 75%** yet still achieves superior results.
>
> |Tasks|PiSSA|LoRA|LoRETTA |
> |-|-|-|-|
> |IC|ViT-Base|**ViT-Base**, **ViT-Large**| ViT-Large|
> |NLU|RoBERTa-Base, **RoBERTa-Large** | RoBERTa-Base, RoBERTa-Large|RoBERTa-Large |
> |NLG|**LLaMA 2-7B**|**LLaMA 2-7B**|LLaMA 2-7B |
>
>
>
> ---
> **R2. The cost of initialization of AdaMSS**
> In our initialization phase, the most time-consuming step is the **estimation of the number of subspaces**.
> As discussed in our ablation study (see Appendix), when the number of subspaces is fixed to $K = 10$, the performance of our method with fixed $K$ is **comparable to AdaMSS with subspace number estimation strategy**.
>
> In the current submitted supplementary material, we further compare AdaMSS **without subspace estimation** to other PEFT methods (including **PiSSA**, **LoRA**, **LoRA-PRO**, and **LoRETTA**) on **LLaMA 2-7B**, **Mistral-7B**, and **Gemma-7B**. In most cases, our method still achieves the **best performance**, even without estimating the number of subspaces.
> To make AdaMSS even more practical, we also apply **low-rank SVD** techniques to accelerate the subspace initialization process.
>
> As shown in the current submitted supplementary material, our method initializes faster than **LoRETTA** and completes in just a few seconds on LLaMA 2-7B, highlighting its computational efficiency.
>
>
> ---
> **R3. Comparison with random initialization**
> We have included a **comparison between random initialization and orthogonal initialization** in the current submitted supplementary material (**Figure 1**). As shown by our experimental results, **orthogonal initialization significantly outperforms random initialization**.
>
> ---
> **R4. Training time/memory comparisons**
> Owing to space constraints, please see our response given in R1 for  Reviewer J6fY.
>
> ---
> **R5. Clarifying generalization and  experiments that demonstrate generalization**
> The generalization ability of a method is reflected in its expected test error. As established in Section **Analysis of AdaMSS**, AdaMSS exhibits a lower upper bound on **the expected test error** compared to other methods, given the same number of training samples. This theoretical advantage is further supported by our experimental results, which consistently show **lower average test errors** for AdaMSS across a range of tasks.
>
> ---
> **R6. Advantages of AdaMSS**   As discussed in  **R1** and **R4–R5**, our method offers the following key advantages.
> - **(1) Superior Accuracy Across Tasks.**
> - **(2) Reduced Training Time and Memory Usage.**
> - **(3) Stronger Theoretical Guarantee:** **Theorem 1** establishes an **upper bound on the expected loss** of AdaMSS, providing a formal **generalization guarantee** that supports its theoretical soundness.
>
> ---
> **R7. Comparison with equivalent trainable parameter counts for LoRA and PiSSA for   language models**
> Thank you so much for the valuable suggestion. Our original comparisons on **ViT models** already included **PiSSA $(r=1)$** as a baseline.
>
> Following your recommendation, we have extended our experiments on **language models** (RoBERTa) to include **LoRA $(r=1)$** and **PiSSA $(r=1)$** under **minimal parameter settings**, in addition to our method **AdaMSS$_{base}$**. Please refer to **Tables 1 and 2** below.
>
> As the experimental results demonstrate, our method consistently outperforms both **LoRA** and **PiSSA** under a   low-parameter budget. Furthermore, **AdaMSS exhibits more stable performance across tasks**, highlighting the advantage of **structured subspace updates** in resource-constrained settings.
>
> **Table 1. Results of LoRA  $(r=1)$, PiSSA $(r=1)$, and AdaMSS$_{base}$ on RoBERTa-Base**
> |Model|Trainable Param|CoLA|MRPC|QNLI|RTE  |SST-2|STS-B|Avg.|
> |-|-|-|-|-|-|-|-|-|
> |LoRA|55K|62.3±3.57|89.22±0.31 |90.59±0.30|**79.49±0.42**|93.74±0.45|80.81±20.64|82.7|
> |PiSSA|55K|62.6±1.44|**89.26±0.61**|90.61±0.41|74.87±1.22|93.28±0.20|89.97±0.28|83.4|
> |AdaMSS|**32K**|**64.5±1.1**|88.8±1.4|**92.4±0.1**|77.3±0.7|**94.6±0.2**|**90.4±0.1**|**84.7**|
>
> **Table 2. Results of LoRA $(r=1)$, PiSSA $(r=1)$, and AdaMSS$_{base}$ on RoBERTa-Large**
> |Model|Trainable Param|CoLA| MRPC|QNLI|RTE|SST-2|STS-B|Avg.|
> |-|-|-|-|-|-|-|-|-|
> |LoRA|147K|62.16±2.38|88.33±0.65|93.86±0.16|82.24±2.47| 95.67±0.41|78.15±29.72|83.4|
> |PiSSA|147K|56.58±6.19|84.90±3.35 |93.40±0.31|65.92±11.26|95.18±0.23|91.26±0.24|81.2|
> |AdaMSS|**45K**|**67.2±1.2**|**90.3±0.5**|**94.5±0.1**|**87.1±2.1**|**96.1± 0.0**|**91.9±0.0**|**87.9**|
>
> ---
> **R8. Add citation [1] and comparison with [1]**
> We sincerely thank the reviewer for bringing this excellent work to our attention. **WeGeFT** is recently accepted method at **ICML 2025**. In the revised version of our manuscript, we have included a discussion of WeGeFT [1] and added experimental comparisons against it.
>
> Below, we provide a summary of  the  differences between **AdaMSS** and **WeGeFT**, as well as empirical comparisons on **Natural Language Understanding (NLU)** and **Image Classification (IC)** tasks.
> - **(1) Motivation:** AdaMSS is motivated by the **multi-subspace structure** of pretrained weight matrices. It seeks compact representations via **lowest-rank representation** and **subspace segmentation**, followed by a **multi-subspace-based adaptive budget allocation** strategy to reduce the number of trainable parameters while preserving model capacity.
>
> In contrast, WeGeFT focuses on **cross-layer parameter sharing**, under the assumption that pretrained weights already contain useful "transferable knowledge" for downstream tasks.
> - **(2) Methodology:** AdaMSS leverages **low-rank representation** and **subspace segmentation** to obtain a compact parameterization of the weights:
>   $\mathbf{W}^{(k)} = \mathbf{W}_0^{(k)} + \mathbf{\hat{W}}_0^{(k)} \Delta \mathbf{Z}^{(k)}$, where $\mathbf{W}^{(k)}$ consists of  the weights   belonging to the $k$-th subspace, $\Delta \mathbf{Z}^{(k)}$ is   **low-rank matrix**, and $\mathbf{\hat{W}}_0^{(k)}$ represents the **principal components** of the corresponding weight subspace.
>
>   WeGeFT adopts the formulation $\mathbf{W}^l = \mathbf{W}_0^{l} + \mathbf{W}_0^{l} \Delta \mathbf{Z}$, where $\Delta \mathbf{Z}$ is a  low-rank matrix shared across layers, and $\mathbf{W}^l$ denotes the weight matrix of the $l$-th layer.
>
> **Comparison with WeGeFT**
>
> **Table 3** compares the performance of AdaMSS and WeGeFT on **NLU** tasks.
> **Table 4** shows their results on **IC** tasks.
> Overall, **AdaMSS outperforms WeGeFT on average** for NLU (RoBERTa) and has a comparable performance for IC (ViT).
>
> **Table 3. Comparison with WeGeFT on NLU (RoBERTa)**
> |Model|Trainable Param |CoLA|MRPC|QNLI|RTE|SST-2|STS-B|Avg.|
> |-|-|-|-|-|-|-|-|-|
> |RoBERTa-Base (WeGeFT)|49K|63.5±1.3| **89.46±0.49**|91.23±0.42|**78.6±1.60**|94.13±0.46|**90.5±0.06**|84.57|
> | RoBERTa-Base (AdaMSS)|**32K**|**64.5±1.1**|88.8 ± 1.4|**92.4±0.1**|77.3±0.7|**94.6±0.2**|90.4±0.1|**84.7**|
> | RoBERTa-Large (WeGeFT)|65K|64.04±1.99|75.74±7.67|93.7±0.3|53.55±1.19|94.95±0.34|91.37±0.26|78.89|
> |RoBERTa-Large(AdaMSS)|**45K**|**67.2 ± 1.2**|**90.3±0.5**|**94.5±0.1**| **87.1±2.1**|**96.1±0.0**|**91.9±0.0**|**87.9**|
>
> **Table 4. Comparison with WeGeFT on IC (ViT)**
> | Model| Trainable Param |CIFAR10|CIFAR100| StanfordCars| OxfordPets| FGVC| EuroSAT| RESISC45|Avg.|
> |-|-|-|-|-|-|-|-|-|-|
> |ViT-Base(WeGeFT)|49K|98.46±0.06|91.46±0.19|76.18±0.20 |92.71±0.15|51.82±0.86|94.92±6.89|93.03±0.19|85.51|
> |ViT-Base(AdaMSS$_{base}$)|**42K**| **98.71±0.07**|**91.90±0.1**|**78.98±0.2**|**93.91±0.2**|**53.2±0.4** |**98.64±0.07**|**93.62±0.08**|**86.99**|
> |ViT-Large (WeGeFT)|**204K**|99.06±0.07 | 92.69±0.19|83.96±0.14|94.49±0.17|60.82±0.46|98.34±0.10|94.49±0.26 |89.12|
> | ViT-Large (AdaMSS)|241K|**99.12±0.00**|**93.22±0.1**|**85.24±0.3**|**94.87±0.1**|**64.31±0.4**|**98.93±0.1**|**94.87±0.1**|**90.13**|
>
> ---
> **R9. Which modules are fine-tuned?**
>
> For both **IC** and **NLU**, we fine-tune **only the Query and Value** projection modules across all methods for a fair comparison. For **Natural Language Generation**, we evaluate two settings: fine-tuning **only the Query and Value** projections, and fine-tuning **all major projection modules**.
>
> ---
>  We thank the reviewer again for the thoughtful questions. Please let us know if any aspects remain unclear—we would be truly grateful for the opportunity to further clarify.

---

> > ### Comment · Reviewer_i4NE · 2025-08-03
> >
> > Thank you for the detailed response. My concerns have been addressed, and I will raise my score from 3 to 4.

---

> > > ### Author Response · Authors · 2025-08-03
> > >
> > > We sincerely thank the reviewer for the kind follow-up and for updating the score. We are truly grateful for the constructive feedback.

---

### Official Review · Reviewer_icwe · 2025-07-02

**Clarity:** 2
**Significance:** 3
**Originality:** 3
**Rating:** 4
**Confidence:** 4

**Summary:**

This paper introduces AdaMSS, a parameter-efficient fine-tuning method that divides neural network weights into multiple subspaces and adapts them with low-rank updates. By dynamically allocating parameters to the most important subspaces, AdaMSS achieves strong performance on vision and language tasks with far fewer trainable parameters than previous methods.

**Questions:**

- The paper claims that lower model complexity leads to better generalization, but this statement is not rigorous enough. Please provide a more detailed explanation.
- Please discuss in more detail how the proposed method compares to other LoRA variants that utilize subspaces.
- How does the proposed method perform in LLM scenarios? Please cover as many base models and parameter scales as possible.

**Ethical Concerns:**

["NO or VERY MINOR ethics concerns only"]

**Final Justification:**

After reading the author's response, some of my concerns have been solved. I keep my previous justification of borderline acceptance.

**Quality:**

3

**Strengths And Weaknesses:**

**Strengths**
- The approach of further reducing LoRA’s trainable parameter count through multiple subspaces while largely maintaining performance is interesting. The introduction of Low-Rank Representation is relatively novel, and the final weight block update strategy is quite clever.
- The paper analyzes the method’s complexity.

**Weaknesses**
- The writing of the paper could be improved, for example by including a method diagram to help readers quickly grasp the main idea. The presentation of experimental results is not intuitive enough, making it hard for readers to immediately see the advantages of the method. The explanation of the importance score in the paper also lacks intuitiveness.
- The paper could include more discussion comparing with other LoRA-based methods that leverage subspaces.

---

> ### Author Rebuttal · Authors · 2025-07-31
>
> We sincerely appreciate the reviewer’s recognition of our work and the constructive feedback and insightful questions provided. Below, we address the concerns and questions raised.
>
> ---
> **R1. Clarity of presentation**
>
> We appreciate the suggestions. Following the recommendations, in the revised version, we will include a **method diagram** to summarize the proposed adaptation process and improve **readability** by simplifying the notation and derivations.
>
> For the experimental section, we will **reorganize the existing ablation studies** and introduce a new **discussion section** to better highlight and explain the advantages of our method.
>
> We believe that these revisions will significantly enhance the **clarity** of our work.
>
> ---
>
> **R2. Intuitive explanation of importance scores**
>
> The **importance score** used in this work is defined based on the product of $\bar{I}^{(t)}$ and $\bar{U}^{(t)}$, where:
>
> - $\bar{I}^{(t)}$   measures the **sensitivity of the weights**, and it is defined on the **magnitude of the gradient-weight product**, which approximates the **change in the loss function** due to that weight.
> - $\bar{U}^{(t)}$ captures the **uncertainty in the sensitivity estimation**.
>
>
> Therefore, this importance score helps identify weights that either (1) contribute significantly to **loss reduction**, or (2) exhibit **uncertainty**, making them important candidates for training.
>
> ---
>
> **R3. Clarifying generalization**
>
> The **superior generalization ability** of a method refers to the following:
> **Given the same number of training samples and comparable training loss**, the method achieves a **lower expected test error** than competing approaches.
>
> To support this, *Theorem 1* provides a generalization bound for **AdaMSS**, showing that the **expected test error decreases with the Gaussian complexity** of the learned function class. Since AdaMSS induces a function class with **lower Gaussian complexity**, it is theoretically expected to generalize better than methods with higher complexity.
>
> This theoretical advantage is further **validated by our empirical results**, which consistently demonstrate that AdaMSS achieves **lower average test errors** across a range of tasks.
>
> ---
> **R4. Discussion in detail how comparing with other LoRA-based subspace methods**
>
> We thank the reviewer for the suggestion to clarify how **AdaMSS** compares to other **LoRA variants that utilize subspaces**. Following your suggestion, we will add a detailed discussion comparing our method with other LoRA-based approaches in the revised version. Besides, we will include a new **Discussion Section** and introduce additional **experiments** to better illustrate the advantages of AdaMSS.
>
> Below is a summary of the discussion for your reference:
>
>
> **1. Key Differences Between AdaMSS and Other Subspace-Based LoRA Variants**
>
> While many existing PEFT methods—such as **PiSSA**, **AdaLoRA** and **LoRA-GA**—rely on the assumption of a **single low-rank subspace**, **AdaMSS** departs from this by modeling the pretrained weight space as composed of **multiple distinct subspaces**.
>
> AdaMSS clusters the column vectors of the weight matrix into separate subgroups, each forming an independent subspace with its own low-rank representation.
> Unlike AdaLoRA, which dynamically adjusts the rank of a single subspace, **AdaMSS performs explicit subspace segmentation and dynamically estimates the importance of each subspace**, enabling more **compact expressiveness** and **finer adaptation**.
>
>
>
> **2. Advantages of AdaMSS over other Methods**
>
> - **(1) Reduced Training Time and Memory Usage:**
>   Thanks to its lower number of trainable parameters and the  proposed adaptive budget allocation, AdaMSS achieves a better performance in both **memory efficiency** and   **training speed**.
>   Moreover, our budget allocation scheme offers greater **flexibility in balancing accuracy and training efficiency**.
>
> - **(2) Better Accuracy Across Multiple Tasks:**
>   AdaMSS consistently outperforms other methods—including **PiSSA**—on several benchmarks, only using **fewer or comparable trainable parameters**.
>   In the following table, we summarize tasks where AdaMSS achieves **at least 3% accuracy improvement** over competing methods. **Bold entries** indicate cases where AdaMSS  **reduces the number of trainable parameters by more than 75%** yet still achieves superior results.
>
> | Tasks | PiSSA | LoRA | LoRETTA |
> |-------|-------|------|----------|
> | IC    | ViT-Base | **ViT-Base**, **ViT-Large** | ViT-Large |
> | NLU   | RoBERTa-Base, **RoBERTa-Large** | RoBERTa-Base, RoBERTa-Large | RoBERTa-Large |
> | NLG   | **LLaMA 2-7B** | **LLaMA 2-7B** | LLaMA 2-7B |
>
>  *In the revised version,  we have extended our experiments on **language models** (RoBERTa) to include **LoRA $(r=1)$** and **PiSSA $(r=1)$** under **minimal parameter settings**, to better highlight and explain the advantages of our method. Please refer to **Tables 1 and 2**.*
>
>
> - **(3) Theoretical Guarantee:**
>   Our method provides a stronger theoretical foundation. **Theorem 1** formally derives an **upper bound on the expected test loss** of AdaMSS, offering a **generalization guarantee** that is not available in most other LoRA-based approaches.
>
>
> **Table 1. Results of LoRA  $(r=1)$, PiSSA $(r=1)$, and AdaMSS$_{base}$ on RoBERTa-Base**
> |Model|Trainable Param|CoLA|MRPC|QNLI|RTE  |SST-2|STS-B|Avg.|
> |-|-|-|-|-|-|-|-|-|
> |LoRA|55K|62.3±3.57|89.22±0.31 |90.59±0.30|**79.49±0.42**|93.74±0.45|80.81±20.64|82.7|
> |PiSSA|55K|62.6±1.44|**89.26±0.61**|90.61±0.41|74.87±1.22|93.28±0.20|89.97±0.28|83.4|
> |AdaMSS|**32K**|**64.5±1.1**|88.8±1.4|**92.4±0.1**|77.3±0.7|**94.6±0.2**|**90.4±0.1**|**84.7**|
>
> **Table 2. Results of LoRA $(r=1)$, PiSSA $(r=1)$, and AdaMSS$_{base}$ on RoBERTa-Large**
> |Model|Trainable Param|CoLA| MRPC|QNLI|RTE|SST-2|STS-B|Avg.|
> |-|-|-|-|-|-|-|-|-|
> |LoRA|147K|62.16±2.38|88.33±0.65|93.86±0.16|82.24±2.47| 95.67±0.41|78.15±29.72|83.4|
> |PiSSA|147K|56.58±6.19|84.90±3.35 |93.40±0.31|65.92±11.26|95.18±0.23|91.26±0.24|81.2|
> |AdaMSS|**45K**|**67.2±1.2**|**90.3±0.5**|**94.5±0.1**|**87.1±2.1**|**96.1± 0.0**|**91.9±0.0**|**87.9**|
>
>
> ---
>
>
>
> **R5. Cover as many base models and parameter scales as possible**
>
> Thank you so much for the comment. Our current experiments cover a diverse set of models, including **ViT (Base/Large)**, **RoBERTa (Base/Large)**, **LLaMA 2-7B**, **Mistral-7B**, and **Gemma-7B**, with parameter sizes ranging from **85.8M to 8.5B**.
> In most cases, our method achieves the **best performance** across these models.
>
> Due to the limited rebuttal timeline, we were unable to include results on larger-scale models in this version. However, we are actively conducting these experiments and will incorporate a more comprehensive comparison with larger-scale models in the revised submission.
>
>
>
> ---
>
> We thank the reviewer again for the thoughtful questions. Please let us know if any aspects remain unclear—we would be truly grateful for the opportunity to further clarify.

---

> > ### Comment · Reviewer_icwe · 2025-08-04
> >
> > Thank you for the author's response, I keep my positive score.

---

> > > ### Author Response · Authors · 2025-08-04
> > >
> > > Thank you so much  for your follow-up and for maintaining a positive score. We are truly grateful for the constructive feedback. We hope our responses have addressed your concerns—please kindly let us know if there’s anything we may have missed or could clarify further.

---

### Official Review · Reviewer_Bhh4 · 2025-07-02

**Clarity:** 3
**Significance:** 3
**Originality:** 3
**Rating:** 5
**Confidence:** 2

**Summary:**

This paper proposes AdaMSS, a parameter-efficient fine-tuning (PEFT) method that adaptively updates a small subset of subspaces within a neural network's weight matrix. Unlike existing methods like LoRA or PiSSA that rely on a single low-rank subspace, AdaMSS identifies multiple subspaces via low-rank representation and performs fine-tuning in a block-diagonal, multi-subspace manner. It also introduces an adaptive budget allocation mechanism that selectively updates only the most relevant subspaces during training. The authors provide theoretical guarantees showing improved generalization bounds over prior PEFT methods. Extensive experiments across vision and language tasks demonstrate that AdaMSS achieves competitive or superior performance with significantly fewer trainable parameters.

**Questions:**

Please check the weakness part. will increase the score if the concerns are solved.

**Ethical Concerns:**

["NO or VERY MINOR ethics concerns only"]

**Final Justification:**

I have read the thorough rebuttal by authors. The paper has strong theoretical analysis with solid performance across boards.

**Limitations:**

Please check the weakness part

**Quality:**

4

**Strengths And Weaknesses:**

# Strengths
1. Insightful empirical finding on multi-subspace structure

The paper presents an interesting and novel empirical observation: pretrained weight matrices across various models (e.g., ViT, RoBERTa, LLaMA) exhibit a multi-subspace structure. This is revealed through Low-Rank Representation (LRR) and spectral clustering, where the coefficient matrix shows an approximate block-diagonal form. This observation serves as a strong motivation for moving beyond single-subspace LoRA-style methods and is well supported by visual and quantitative evidence.

2. Well-motivated Multi-Subspace-Based Incremental Update

The proposed AdaMSS framework is a conceptually clean and practically relevant solution. By projecting pretrained weights into multiple orthogonal subspaces and assigning fine-tuning capacity selectively based on subspace importance, the method provides a structured, task-aware approach to parameter-efficient fine-tuning. The use of residual connections further ensures expressiveness while maintaining modularity.

3. Strong theoretical analysis

The paper is theoretically grounded, providing clear generalization bounds via Gaussian complexity analysis. The bounds support the core claim that multi-subspace adaptation leads to improved generalization compared to global low-rank methods. This theoretical support enhances the credibility and rigor of the proposed method.

4. Solid empirical performance across domains

AdaMSS demonstrates strong and consistent performance improvements over existing PEFT methods such as LoRA, PiSSA, and RandLoRA across a wide range of benchmarks. Notably, it achieves these gains with significantly fewer trainable parameters, which makes it especially suitable for large-scale or resource-constrained applications.

# Weaknesses
1. Sufficiency of empirical evidence for subspace structure

Figure 2 is a good illustrative starting point, but on its own, it's not sufficient evidence to claim that pretrained models broadly exhibit a multi-subspace structure. More layer-wise or task-relevant analysis would be needed to make the case compelling.

2. Lack of comparative baselines for adaptive subspace selection

The proposed adaptive budget allocation strategy, which freezes or updates subspaces based on gradient magnitude and variance, is intuitive. However, it would be useful to see direct comparisons with simpler or alternative selection strategies, such as random subspace selection or allocating budget based on principal singular directions within each cluster. This would help isolate the specific benefit of the adaptive mechanism.

3. Sensitivity to preprocessing and hyperparameters

The method introduces several preprocessing steps and hyperparameters, such as the number of subspaces
𝐾 clustering thresholds, and gradient-based pruning schedules. It remains unclear how sensitive the method is to these choices or how automated the process is in practice. More guidance or ablation studies would be helpful, especially for practitioners looking to apply the method to new domains.

## Writing Clarity
The paper can be somewhat difficult to follow in parts. The writing is dense, particularly in Sections 3.1 and 3.2, where the mathematical derivation of subspace segmentation and reparameterization could be broken down more clearly. The interplay between different components is not always intuitive on first read. More visual aids or examples could significantly improve clarity for readers unfamiliar with subspace learning techniques.

---

> ### Author Rebuttal · Authors · 2025-07-31
>
> We sincerely appreciate the reviewer’s recognition of our work and the constructive feedback and insightful questions provided. Below, we address the concerns and questions raised.
>
> ---
> **R1. Sufficiency of empirical evidence for subspace structure**
> Thank you so much for your constructive suggestions. In addition to **Figure 2**, we will include **more visualizations of empirical evidence for multi-subspace structures across layers and tasks** in the revised manuscript. Owing to space constraints, we  provide **partial numerical evidence** for multi-subspace structures across different layers.
>
> In addition to the **approximate block-diagonal patterns** shown in Figure 2, another way to observe the distribution of weight column vectors is through the **singular values of the Laplacian matrix** constructed from the **principal components**. In **Table 1**, we show the singular value distribution of the Laplacian matrix computed from the principal components of the **query weight matrices** at each layer of a pretrained **`ViT-Large`** model. From Table 1, we make the following observation: for most layers, the **first 900 singular values remain significantly large**, while the trailing values are **very close to zero** (e.g., below 0.01).
>
> The **number of near-zero singular values** of the Laplacian matrix can be interpreted as an estimate of the **number of disjoint subspaces** within the weight space. This provides an additional numerical evidence of the **multi-subspace structures** within the principal components of pretrained weights.
>
>
>
> **Table 1: Singular value distribution of the Laplacian matrices constructed from the principal components of the query weight matrices at each layer of the pretrained `ViT-Large` model.**
> |Layer index|$\sigma_1$|$\sigma_{101}$|$\sigma_{201}$|$\sigma_{301}$|$\sigma_{401}$|$\sigma_{501}$|$\sigma_{601}$|$\sigma_{701}$|$\sigma_{801}$|$\sigma_{901}$| $\sigma_{1001}$ | $\sigma_{1024}$|
> |-|-|-|-|-|-|-|-|-|-|-|-|-|
> |1|1.00|0.87|0.78|0.59|0.38|0.21|0.09|0.02|0.00|0.00|0.00|0.00|
> |3|1.00|0.83|0.78|0.72|0.65|0.57|0.47|0.37|0.24|0.10|0.00|0.00|
> |5|1.00|0.83|0.81|0.79|0.77|0.74|0.70|0.64|0.54|0.38|0.05|0.00|
> |7|1.00|0.82|0.81|0.79|0.77|0.74|0.71|0.67|0.60|0.47|0.01|0.00|
> |9|1.00|0.82|0.81|0.79|0.78|0.76|0.74|0.71|0.67|0.56|0.00|0.00|
> |11|1.00|0.83|0.81|0.80|0.79|0.77|0.75|0.73|0.68|0.55|0.00|0.00|
> |13|1.00|0.83|0.81|0.80|0.78|0.76|0.74|0.71|0.66|0.52|0.00|0.00|
> |15|0.99|0.81|0.79|0.77|0.76|0.73|0.71|0.67|0.63|0.53|0.00|0.00|
> |17|1.00|0.82|0.80|0.79|0.77|0.76|0.74|0.71|0.68|0.60|0.00|0.00|
> |19|1.00|0.82|0.81|0.80|0.78|0.77|0.76|0.74|0.71|0.66|0.00|0.00|
> |21|1.00|0.82|0.81|0.80|0.80|0.79|0.78|0.77|0.75|0.72|0.00|0.00|
> |23|1.00|0.83|0.82|0.81|0.80|0.80|0.79|0.78|0.77|0.75|0.12|0.00|
>
> ---
>
> **R2. Lack of comparative baselines for adaptive subspace selection**
>
> **Table 2:   Comparing the performance of different adaptive allocation scheme on `ViT-Large` model.**
> | Methods \ Dataset | StanfordCars | CIFAR100 | FGVC |
> |-|-|-|-|
> |  Importance-score-based  adaptive budget allocation |**85.24±0.3**|**93.22±0.01**|**64.31±0.4**|
> | Random adaptive selection   | 83.63±0.30 | 93.11±0.10 | 59.36±0.72 |
> | Adaptive selection based on $\ell_1$ norm of each subspace parameters  | 79.89±0.41 | 92.00±0.21 | 40.77±1.13 |
>
>
> Thank you so much for your constructive suggestions. We agree that including comparisons against alternative strategies will help clarify the advantages of our adaptive allocation scheme.
>
> In the revised manuscript, we include comparisons with the following two adaptive allocation strategies:
> - (1) **Random adaptive selection**, as suggested by the reviewer.
> - (2) **$\ell_1$-norm-based adaptive budget allocation**, where the budget is assigned based on subspaces with the highest $\ell_1$-norm, although we currently do not directly implement allocation based on **principal singular directions within each cluster** as suggested.
>
> As shown in **Table 2**, $\ell_1$-norm-based adaptive budget allocation and random adaptive selection  lead to significant performance degradation, and importance-score-based  adaptive budget allocation has achieved best performance among all compared methods.
>
> ---
> **R3. Sensitivity to hyperparameters and preprocessing**
>
> Thank you so much for your constructive suggestions. In the current version of the manuscript, we have already compared strategies **with and without subspace number estimation**, as well as the effect of using different fixed values of $K$.
>
> Following your suggestion, we have reorganized our existing ablation studies and added **new ablations** from the following two aspects:
>
> 1. **Hyperparameters for subspace segmentation**, including the thresholds $\tau$ and $K_0$ used for estimating the number of subspaces $K$;
> 2. **Hyperparameters for adaptive budget allocation**, including the target number of subspaces $K_{\text{target}}$ and the decay exponent $\rho$ used in the proposed multi-subspace adaptive budget allocation scheme.
>
> Below we summarize the findings of these new ablation studies:
>
> - **(1) Thresholds $\tau$ and $K_0$ for subspace estimation**:
>   The threshold $\tau$ is used to detect near-zero singular values in the normalized Laplacian matrix and is set as a small value. In our experiments, we set $\tau = 0.01$. The parameter $K_0$ serves as a lower bound on the estimated number of subspaces.
>   In **Tables 3–5**, we evaluate various values $\tau$ ∈ {0.001, 0.01, 0.05, 0.10, 0.15, 0.20} and $K_0$ ∈ {1, 5, 10, 15, 20} using **ViT-Large** with  AdaMSS$_{base}$ (without adaptive budget allocation).
>
>    The results show that when $K_0 \geq 10$,  AdaMSS$_{base}$ exhibits robust  performance  across all three datasets for  different  $\tau$ and $K_0$.
>
> - **(2) Target number of subspaces $K_{\text{target}}$ and decay exponent $\rho$**:
>   In the adaptive budget allocation scheme, the number of trainable subspaces is gradually reduced to $K_{\text{target}}$ according to a **smooth cubic decay schedule** (default $\rho = 3$). A larger $\rho$ indicates a faster decay in the number of active subspaces.
>   In **Tables 6–8**, we evaluate the sensitivity to $\rho$ ∈ {1, 2, 3, 4, 5} and $K_{target}$ ∈ {100, 200, 300, 400} using **ViT-Large** with  **AdaMSS**.
>   The results indicate that **AdaMSS is robust** to across a range of $\rho$ and $K_{\text{target}}$.
>
>
> **Table 3: Results of AdaMSS**$_{base}$ **under different hyperparameter settings for `StanfordCars`.**
> | $\tau$ \ $K_0$|1|5|10|15|20|
> |-|-|-|-|-|-|
> |0.001|84.25±0.23|85.40±0.19|85.40±0.22|85.38±0.25|85.67±0.42|
> |0.01|84.12±0.24|85.20±0.50|85.31±0.27|85.41±0.13|85.50±0.21|
> |0.05|84.77±0.16|85.02±0.26|85.23±0.28|85.59±0.27|85.70±0.24|
> |0.10|84.58±0.29|84.93±0.25|85.60±0.26|85.36±0.29|85.42±0.23|
> |0.15|84.72±0.39|85.12±0.23|85.38±0.33|85.61±0.20|85.63±0.45|
> |0.20|84.57±0.09|84.86±0.17|85.53±0.21|85.39±0.34|85.65±0.19|
>
> **Table 4: Results of AdaMSS$_{base}$ under different hyperparameter settings for `CIFAR100`.**
> | $\tau$ \ $K_0$|1|5|10|15|20|
> |-|-|-|-|-|-|
> |0.001|93.27±0.09|93.44±0.16|93.47±0.06|93.57±0.12|93.48±0.15|
> |0.01| 93.38±0.08|93.33±0.10|93.50±0.10|93.53±0.15|93.57±0.05|
> |0.05|93.42±0.12|93.43±0.12|93.51±0.09|93.45±0.10|93.56±0.09|
> |0.10|93.40±0.12|93.47±0.09|93.58±0.13|93.45±0.12|93.64±0.10|
> |0.15|93.43±0.12|93.43±0.14|93.52±0.06|93.55±0.06|93.60±0.12|
> |0.20|93.43±0.10|93.54±0.05|93.48±0.08|93.53±0.04|93.51±0.11|
>
> **Table 5: Results of AdaMSS$_{base}$ under different hyperparameter settings for `FGVC`.**
> |$\tau$ \ $K_0$|1|5|10|15|20|
> |-|-|-|-|-|-|
> |0.001|61.36±1.12|65.33±0.55|66.11±0.53|66.62±0.78|66.98±0.73|
> |0.01|61.90±0.55|64.58±0.69|65.27±0.64|66.10±0.69|66.60±0.68|
> |0.05|61.64±0.22|64.63±0.98|65.51±1.05|66.63±0.48|67.44±0.68|
> |0.10|62.42±0.33|64.49±0.87|65.88±0.68|66.29±0.21|67.16±0.82|
> |0.15|62.56±0.73|64.90±1.02|65.72±1.04|66.86±0.30|66.44±0.92|
> |0.20|62.00±0.51|64.76±0.78|65.68±0.93|66.65±0.41|66.86±0.60|
>
> **Table 6: Results of AdaMSS under different hyperparameter settings for `StanfordCars`.**
>
> |$\rho$ \ $K_{target}$|100|200|300|400|500|
> |-|-|-|-|-|-|
> |1|84.78±0.32|85.06±0.16|85.34±0.13|85.35±0.11|85.16±0.12|
> |2|84.88±0.36|85.11±0.19|85.21±0.33|85.26±0.20|85.21±0.41|
> |3|85.03±0.29|84.91±0.24|85.21±0.39|85.06±0.20|85.02±0.18|
> |4|84.69±0.20|84.95±0.13|85.03±0.29|85.02±0.21|85.19±0.13|
> |5|84.89±0.27|84.67±0.25|85.04±0.25|85.16±0.26|85.37±0.36|
>
> **Table 7: Results of AdaMSS under different hyperparameter settings for `CIFAR100`.**
> |$\rho$ \ $K_{target}$|100|200|300|400|500|
> |-|-|-|-|-|-|
> |1|93.47±0.10|93.49±0.11|93.41±0.14|93.56±0.10|93.34±0.12|
> |2|93.41±0.14|93.44±0.07|93.54±0.08|93.42±0.13|93.47±0.07|
> |3|93.42±0.14|93.38±0.05|93.56±0.09|93.46±0.14|93.55±0.08|
> |4|93.25±0.11|93.53±0.09|93.40±0.10|93.48±0.15|93.50±0.07|
> |5|93.35±0.15|93.45±0.11|93.37±0.09|93.41±0.12|93.41±0.12|
>
> **Table 8: Results of AdaMSS under different hyperparameter settings for `FGVC`.**
> |$\rho$ \ $K_{target}$|100|200|300|400|500|
> |-|-|-|-|-|-|
> |1|64.76±0.55|65.56±0.60|64.98±0.81|64.90±0.98|65.73±0.48|
> |2|64.55±0.73|64.38±0.83|65.14±0.89|65.11±0.57|64.99±0.70|
> |3|63.73±0.31|64.16±1.22|64.77±0.34|65.33±0.64|65.77±0.21|
> |4|63.02±0.89|64.35±0.62|64.64±0.81|64.54±0.47|65.13±0.61|
> |5|62.74±1.53|63.43±0.63|65.12±0.83|64.28±0.99|65.32±0.73|
>
> ---
> **R4. Writing clarity and visual explanation**
>
> Thank you for highlighting this issue. We will revise **Sections 3.1 and 3.2** by simplifying the notation and derivations, and by adding **visual illustrations** for the proposed methods.
> We hope these revisions will enhance the **clarity** and **readability** of our work.
>
> ---
> We thank the reviewer again for the thoughtful questions. Please let us know if any aspects remain unclear—we would be truly grateful for the opportunity to further clarify.

---

> > ### Comment · Reviewer_Bhh4 · 2025-08-05
> >
> > Thanks for the rebuttal which solved my concerns. I will increase my score.

---

> > > ### Author Response · Authors · 2025-08-05
> > >
> > > We sincerely thank the reviewer for the kind follow-up and for increasing the score. We are truly grateful for the constructive feedback.

---

### Official Review · Reviewer_J6fY · 2025-07-04

**Clarity:** 3
**Significance:** 3
**Originality:** 3
**Rating:** 5
**Confidence:** 2

**Summary:**

The paper introduces AdaMSS, a new parameter-efficient fine-tuning (PEFT) method for large language models. Rather than using fixed low-rank adapters (like LoRA), AdaMSS selects a set of low-rank subspaces adaptively with a learned budget controller. The method is supported by solid theoretical analysis and evaluated across NLP and vision tasks. It shows promise in reducing trainable parameters while maintaining performance.

**Questions:**

- How well can this method scales with long-context or multi-step reasoning tasks?
- Can it be extended with positional tuning approaches like RoCoFT?

**Ethical Concerns:**

["NO or VERY MINOR ethics concerns only"]

**Final Justification:**

The paper presents a novel idea with a good theoretical foundation. The authors responded and addressed my concerns. They plan to provide a revised version with the additional results and discussion.

**Limitations:**

Yes

**Quality:**

3

**Strengths And Weaknesses:**

Strengths:

- The paper introduces a novel idea and provides a strong theoretical foundation.
- It provides an extensive experimental study to demonstrate the viability of the approach across multiple domains, including LLMs and vision transformers.
- The approach achieves significant parameter efficiency while maintaining high performance comparable to SOTA methods.

Weaknesses:

- The paper does not include performance metrics such as GPU memory use or training speed, which are key for PEFT.
- Some of the newer models, such as  LLaMA3, DeepSeek, or Gemini, are not included in the experimental study.

---

> ### Author Rebuttal · Authors · 2025-07-31
>
> We sincerely appreciate the reviewer’s recognition of our work and the constructive feedback and insightful questions provided. Below, we address the concerns and questions raised.
>
> ---
>
> **R1. Performance metrics such as GPU memory or training speed**
>
>
> Thank you very much for pointing this out. In the revised version, we have added a comparison of different PEFT methods in terms of both memory usage and training time. Since our method use a smaller number of trainable parameters, it demonstrates superior memory efficiency, as shown in **Tables 1 and 2**.
>
> Regarding training speed, as discussed in the paper, the computational complexity of our gradient updates is of the same order as that of  LoRA and PiSSA when $r = \sum_{k=1}^{K} r_k$, where $r$ denotes the hyperparameter used in **LoRA** and **PiSSA**. However, in practice, the multi-subspace-based adaptive budget allocation allows for faster training by selectively updating only the most important subspaces (see **Tables 3 and 4**).
>
>
> Besides, the proposed   multi-subspace-based adaptive budget allocation provides greater flexibility in balancing training time and performance. In our paper, the number of trainable subspaces is gradually reduced to a target value $K_{target}$ following a smooth  decay schedule  with decay exponent $\rho=3$. A larger $\rho$ leads to a faster reduction in the number of trainable subspaces. **Table 5** presents an ablation study demonstrating how different choices of $\rho$ affect training speed and final performance.
>
>
>  **Table 1: Optimizer memory consumption (MB) of different PEFT methods on `ViT-Large` using fp32 precision.**
>  | Method        | LoRA $(r=16)$  | PiSSA $(r=8)$ | LoRETTA $(r=5)$ |AdaMSS$_{base}$ $(r_k=1)$|
>  |---------------|------- |------- |--------|------|
>  |Memory (MB) |  18.84 |  9.6 |  1.584  |  2.136 |
>
>
> **Table 2: Optimizer memory consumption (MB) of different PEFT methods on `Roberta-Large` using fp32 precision.**
> | Method        | LoRA $(r=8)$  | PiSSA $(r=8)$ | LoRETTA $(r=5)$ |AdaMSS $(r_k=1)$|
> |---------------|------- |------- |--------|------|
> |Memory (MB) | 9.6 | 9.6 | 1.58  | 0.54 |
>
>
> **Table 3: Average training time (in seconds) of different PEFT methods on `ViT-Large`.**
> | Datasets\Methods       | LoRA  | PiSSA | LoRETTA |AdaMSS $(r_k=3, \sum_{k=1}^{K} r_k\geq 30)$|
> |---------------|------- |------- |--------|------|
> | FGVC  | 1207.7669| 1206.7017  |  1057.7328  | 1074.9871 |
>
>
>
> **Table 4: Average training time (in seconds) of different PEFT methods on  `Roberta-Large` for  natural language understanding task (STS-B).**
> | Method        | LoRA $(r=8)$ | PiSSA $(r=8)$ | LoRETTA $(r=5)$ |AdaMSS $(r_k=1, \sum_{k=1}^{K} r_k\geq 10)$ |
> |---------------|------- |------- |--------|------|
> | Avg. Time (s) |5325.77 | 5174.04 | 5441.47  | 4972.03|
>
> **Table 5: Training time (in seconds) and performance (PCC) of AdaMSS with varying $\rho$ on  `Roberta-Large` for natural language understanding task (STS-B).**
> |   $\rho$    | 5  | 10 | 15 |
> |---------------|------- |------- |--------|
> | Avg. Time (s) |4913.63 | 4780.97 | 4667.35 |
> |  PCC   | 91.66±0.03 | 91.52±0.02 | 91.35 ± 0.03 |
>
>
> ---
> **R2. Experimental study on newer models (LLaMA3, DeepSeek, Gemini)**
>
> Thank you  so much for the valuable comments. Our current experiments cover a range of models, including **ViT (Base/Large)**, **RoBERTa (Base/Large)**, **LLaMA 2-7B**, **Mistral-7B**, and **Gemma-7B**. Among them, **LLaMA 2-7B** and **Gemma-7B** were released in *2023*, while **Mistral-7B** was released in *2024*.
>
> Due to the time constraints of the rebuttal, we were unable to include results on **LLaMA 3** and **DeepSeek** in this version. However, we are actively working on these experiments and will include a more comprehensive comparison with newer models in the revised version.
>
> ---
> **R3. Scalability to long-context or multi-step reasoning tasks**
>
> Thank you for the question. Our method is scalable to long-context and multi-step reasoning tasks. This is supported by our experimental results on **GSM8K** and **MATH** — two widely recognized benchmarks for multi-step symbolic and numerical reasoning. Compared to other methods, our approach achieves **higher accuracy** while requiring **fewer trainable parameters**, demonstrating both its **efficiency** and **effectiveness** in handling complex reasoning scenarios. The detailed results are provided below for reference.
>
> **Table 6: Perfromance of diferent PEFT methods on LLaMA 2-7B.**
>  | Method        | Trainable Parameters | GSM8K   | MATH         |
> | -------------------- | -------------------- | ------------- | ------------ |
>  | Full FT              | 6738M                | 49.05 | 7.22 |
>  | LoRA                | 320M                 | 42.30         | 5.50         |
> | PiSSA $(r=8)$         | 19M                  | 44.11         | 5.84         |
> | LoRA-PRO $(r=8)$     | 19M                  | 46.61         | 6.40         |
> | AdaMSS$_{base}$ $(r_k=3)$ | **4M**               | **51.10**     | **7.57**     |
> | AdaMSS$(r_k=3)$    | **4M**               | 50.80     | 7.22     |
>
>
> **Table 7: Perfromance of diferent PEFT methods on Mistral-7B.**
> | Method               | Trainable Parameters | GSM8K     | MATH      |
> | -------------------- | -------------------- | --------- | --------- |
> | Full FT | 7242M | 67.02  | 18.60   |
> | LoRA   | 168M    | 67.70     |  19.68  |
> | PiSSA $(r=8)$  | 20M |  71.00  |  20.40 |
> | LoRA-PRO $(r=8)$ | 20M | 69.59| 19.17 |
> | AdaMSS$_{base}$$(r_k=3)$ | 4M |**70.71**|**20.44** |
> | AdaMSS$(r_k=3)$| **2M** |  70.74  | 19.47|
>
>
> **Table 8: Perfromance of diferent PEFT methods on Gemma-7B.**
> | Method               | Trainable Parameters | GSM8K     | MATH      |
> | -------------------- | -------------------- | --------- | --------- |
> | Full FT | 8538M | 71.34 | 22.74|
> | LoRA | 200M  |74.90 |31.28|
> | PiSSA $(r=8)$ | 25M | 75.48 | 29.59 |
> | LoRA-PRO  | 25M | 75.90 | 29.25 |
> | AdaMSS$_{base}$ $(r_k=3)$ | 6M     | 75.33     | **29.73** |
> | AdaMSS $(r_k=3)$ | **4M**  | **76.41** | 28.64     |
>
>
> ---
> **R4. Compatibility with positional tuning (e.g., RoCoFT)**
>
>
> We thank the reviewer for this insightful question. Indeed, our framework could potentially be integrated with positional tuning methods such as **RoCoFT**.
>
> One preliminary yet intuitive idea is to apply positional tuning directly to the **low-rank representation** $\mathbf{Z}$, especially given that $\mathbf{Z}$ often exhibits an approximately **block-diagonal structure**. Another promising direction, inspired by the **AdaMSS**, is to leverage **subspace segmentation** to guide RoCoFT in deciding **which columns should be updated and which can be frozen**, potentially enabling **adaptive budget allocation** during training.
>
> We will include this discussion in the *Future Work* section of the revised version.
>
>
>
>
> ---
> We thank the reviewer again for the thoughtful questions. Please let us know if any aspects remain unclear—we would be truly grateful for the opportunity to further clarify.

---

> ### Author Response · Authors · 2025-08-07
>
> Dear Reviewer J6fY,
>
> Thank you again for your valuable feedback on our submission.
>
> We have submitted a detailed rebuttal addressing your comments, including:
>
> - a comparison of **training time and memory usage** across different methods **(Tables 1–5)**;
> - an update on **newer models**, including coverage of `Mistral-7B (2024)` and `Gemma-7B (2024)` **(Tables 7–8)**, with additional comparisons (e.g., `LLaMA 3 (2024)` and `DeepSeek`) currently in progress for inclusion in the revised version;
> - a discussion on **scalability** to long-context and multi-step reasoning tasks **(Tables 6–8)**;
> - a discussion on **compatibility with positional tuning approaches**.
>
> Your recognition would mean a lot to us.
>
> Best regards,
> The Authors

---

> ### Comment · Area_Chair_iJTY · 2025-08-08
>
> Hi Reviewer J6fY,
>
> The authors have provided a detailed response; do they sufficiently address your concerns on baseline comparison and computational efficiency?
>
> Best,
> AC

---

### Note · Authors · 2025-08-12

Dear Area Chairs and Reviewers,

We sincerely appreciate the time and effort you have dedicated to handling and reviewing our submission. We are also grateful for the positive feedback from all reviewers, as well as for the constructive suggestions provided to improve the work.

Although comments provided by reviewers after the author–reviewer discussion period are no longer visible to us, we still wish to express our gratitude for any additional feedback and for the valuable time you have invested. We believe that your suggestions will be highly beneficial in further improving this work.

Best,

The Authors

---

### Decision · Program_Chairs · 2025-09-17

**Decision:**

Accept (poster)

**Comment:**

AdaMSS is a novel PEFT method that segments pretrained weights into multiple low-rank subspaces, overcoming the limited expressiveness of single-subspace approaches. This multi-subspace design enables richer adaptation without significantly increasing trainable parameters, striking a better balance between expressiveness and efficiency. The method further introduces adaptive freezing to focus updates on the most important subspaces and provides a theoretical generalization bound with lower Gaussian complexity. Experiments across vision and NLP tasks confirm strong gains over existing PEFT methods while using fewer parameters.

The rebuttal addressed concerns with added efficiency metrics, expanded evaluation (Mistral-7B, Gemma-7B), stronger ablations, and clarified theory. While exposition could be clearer and results on the very latest models are pending, the contributions are timely, impactful, and well-supported, making the work valuable to the PEFT and efficient LLM communities and likely to inspire further research on structured subspace-based fine-tuning.